# High-Performance Machine Learning for FinTech

## Abstract

This paper introduces a high-performance compute engine based on differential evolution and tailored for machine learning techniques in the FinTech sector. We demonstrate enhanced runtime performance, which allows for testing a broader array of candidate investment and trading strategies, thereby expanding the scope and improving the quality of strategy evaluations. Serving as the foundation of our differential-evolution-based machine learning framework for portfolio selection and management, this engine is designed for processing real market data and executing highly customisable trading strategies. We present various techniques to optimise its runtime performance, assess their relative impact on performance, and quantify its superior performance compared to existing engines.

## 1 Introduction

In the FinTech domain, successful machine learning engines rely on robust computational frameworks capable of assessing diverse potential investment and trading strategies efficiently. Central to this setup is the use of modern AI and ML methods, such as neural networks and genetic programming. These methods excel in testing numerous strategies and models, but their flexibility adds to the computational burden by increasing the number of strategy variations to be assessed due to different parametrisations and tuning procedures. This exacerbates the challenges that data analysts face in managing large data volumes and assessing an increasingly larger universe of complex strategies. We address these two challenges by developing a highly optimised and efficient compute engine and defining a rule-based expression language. This paper focuses on the first challenge by detailing the intricacies of the efficiency enhancements made to a compute engine that we developed but references essential aspects of the rule-based expression language that we designed and tailored for FinTech applications, and which we present in a separate publication (<anonymised> (2022b)).

This paper is about constructing a machine learning (ML) system tailored for FinTech, addressing performance challenges associated with handling large data volumes and backtesting. We present a structured analysis and evaluation of our compute engine's computational efficiency, contrasting it with other extant platforms offering ML capabilities for FinTech. By dissecting the engine's core architecture, we suggest various optimisation methods to enhance efficiency and assess their impact across the engine's components on overall performance. Such identification of key efficiency enhancing factors offers guidance to developers on optimising machine-learning engines. The main contributions of this paper relate to the specific techniques devised to enhance computational efficiency in the different components of the engine and to an empirical analysis of the engine's performance in a longitudinal study production setting. To put these contributions in context, we also summarise the contributions made in <anonymised> (2022b) with respect to the design features of the developed expression language that allow a high level of automation.

The remainder of the paper is organised as follows.

Section 2 presents the problem's context and background through a review of the relevant literature and features of comparative extant platforms.

Section 3 summarises the main contributions made in the design of the expression language that are relevant to the analysis in this paper. This design facilitates high-level automation for empirical research on strategy development in FinTech. It automates more decisions for investment managers and traders than extant tools, enhancing decision-making by replacing subjectivity with data-driven insights. This is not meant to replace

professionals but to equip them with tools and techniques for making informed decisions about strategy robustness over time.

Section 4 describes the techniques devised and used to enhance the speed of backtesting engines that are the main focus of this paper. A high-performance engine is essential for conducting computationally intensive simulations necessary for high-level decision-making that utilise AI and ML. Our thorough exploration of these techniques distinguishes our approach from others, presenting a significant contribution to professional practice and computational research in backtesting engines. These techniques also hold promise for enhancing empirical research in time series analysis and financial studies across other research tools and ML engines.

Section 5 conducts an empirical analysis to assess the tool's speed for longitudinal studies and portfolio rebalancing between trading sessions. A summary of performance results confirms the viability of both tasks. This is provided as a brief test in a production setting, but this paper focuses solely on computational discussions and tool comparisons.

Finally, Section 6 provides a summary and insights into implications for future financial research.

## 2 Context and Background

**Research Gap:** Previous studies, such as Kumiega & Van Vliet (2012), have identified a gap in research concerning the development processes of algorithmic trading strategies. Algorithmic trading has gained prominence in financial markets, shifting the focus from individual trader decisions to long-term managerial decisions that rely on trading infrastructure and automation tools. Key considerations include which trading and investment strategies warrant further exploration for profitability and robustness, and which require refinement for live trading, encompassing execution, user interface enhancements, and automation of specialised scenarios. Developing specific strategies or features, however, can tie up developer resources for extended periods. Without safe, extensive, and flexible means to assess decisions at this level, individuals may form biased estimates of success probabilities, potentially influenced by cognitive or data mining biases or erroneous assumptions in backtesting. Thus, a significant contribution can be made by providing development managers and researchers with safe, capable, and flexible automated tools to test the robustness of encoded processes, where financial theories are made machine-executable, testable, and their results quantifiable. Failure to do so may lead to inefficient allocation of resources and unreliable returns. Enhancing this process can facilitate efficiency gains in money and time. This is what our compute engine aims to achieve.

**Literature:** Machine learning has gained significant traction in both business and research due to its ability to detect patterns in vast datasets, which previously required extensive manual analysis by data experts. Rather than replacing data analysts, ML has augmented their capabilities through innovative techniques driven by computational advancements. Evolutionary computing methods, as highlighted by Bose & Mahapatra (2001), are recognised for their effectiveness in identifying patterns in noisy data across extensive problem spaces, particularly when the data format is consistent, as is often the case in finance. Despite the regulatory and acceptance hurdles faced by ML (Lui & Lamb, 2018), its transparency and comprehensibility have made it a preferred choice in finance (Wall, 2018). In parallel, genetic programming generates human-readable expressions and is deemed suitable for strategy development simulation (Lohpetch, 2011). A plethora of other techniques such as deep neural networks, decision trees, clustering, classification, support vector machines, and hidden Markov models also augment the set of available techniques (Bao et al., 2017; Wu et al., 2006; Aronson & Masters, 2013; Luo & Chen, 2013; Hargreaves & Mani, 2015; Nguyen, 2018; Arnott et al., 2019). These techniques stand to benefit from enhanced simulation speeds and can be amalgamated with genetic programming to forge hybrid ML methodologies. Genetic programming can leverage inputs from technical analysis (Colby, 2002) or features derived from other ML algorithms, and can also be employed for meta-optimisation of other ML techniques. This underpins our initial focus on genetic programming and the formulation of an expression language apt for modelling hybrid ML strategies.

Inspired by AutoML capabilities for making ML methods and processes available for non-ML experts Kotthoff et al. (2017), our research aims to develop analogous tools tailored for time-series analysis within the financial trading realm. While our primary focus lies within this specific domain during the developmental phase, the methodologies and solutions we devise are transferable across diverse research domains.

**Robustness Problems:** ML and manual strategy development in financial markets face challenges due to transient exploitable inefficiencies and potential biases in detecting them. Aronson (2011) identifies biases like data-mining, selection, survivorship, and curve-fitting biases and proposes mitigating scientific approaches. Given the fast evolving nature of markets, strategies must adapt quickly to ensure their currency and robustness. We follow Pardo (2011) who advocates for a walk-forward automation approach, akin to an automated longitudinal study, to validate strategy robustness and mitigate potential biases. To this end, high-speed automation tools are essential for conducting such extensive simulations and empirical research to assess strategy development processes effectively.

**Existing Solutions:** We initially examined the literature on platform choices for back-testing in financial research. Options range from general-purpose scripting languages like Python (de Prado, 2018; Jansen, 2020) and Julia (Danielsson, 2011), to statistical platforms such as Matlab (Chan, 2017) and R (Georgakopoulos, 2015; Conlan, 2016), as well as specialised trading and FinTech platforms like TradeStation (Pardo, 2011; Ehlers, 2013), MetaTrader (Blackledge et al., 2011), NinjaTrader (Ford, 2008), FXCM Trading Station (Mahajan et al., 2021), and Zorro (Liang et al., 2020).

**Technical Focus:** We focus primarily on event-driven backtesting engines while contrasting some vector-based ones. However, vector-based strategies often require rewriting for live trading, pose challenges in aligning multiple data feeds, and are susceptible to the 'look-ahead' bias during research.[1] Performance comparisons among Matlab, R, Python, and Java have already been performed in JuliaLangContributors (2022). The Julia benchmarks might be biased positively since Python can be faster than Julia in some cases (Vergel Eleuterio & Thukral, 2019). The Java benchmarks are biased negatively as they are influenced by Java Native Interface (JNI) integration for C-based matrix libraries. JNI adds significant overhead to function calls and conducted tests mainly measure the performance of the C-binding of the matrix library. This is adequate when comparing matrix operations where Matlab, R, Python, and Julia have their strengths, but they may not be the best-performing solutions for financial backtests. While these general-purpose languages and trading platforms lack ready-made strategy generation solutions, specialist platforms like StrategyQuantX and Adaptrade (Hafiiak et al., 2018; Hayes et al., 2013) and commercial tools like GeneticSystemBuilder (Zwag, 2020) and BuildAlpha (Bergstrom, 2020) offer faster alternatives.

**Contribution:** These existing tools, both commercial and non-commercial, lack the computational performance and automation capabilities needed for robustness testing in trading and investment strategy development processes. In the following sections, we address these research gaps in our high-performance backtesting engine. This advancement enhances professional practice and empirical research by enabling cost-effective scenario testing in terms of processing power, manual labour, and time. Our domain-specific expression language and highly optimised computing engine facilitate the exploration of cutting-edge machine learning technologies in algorithmic trading and portfolio management.

## 3 A High-Performance Platform: Invesdwin

To improve the current capabilities for empirical research on strategy development processes in finance, the first author has implemented a novel platform in Java. This platform, dubbed 'Invesdwin', has been extended over the years to enable high-performance machine learning with a high level of automation. This section outlines how users can analyse strategy development processes using the platform and discusses key architectural and design features that ensure high performance.

Figure 1 outlines the key structural features and processes of the platform, with a focus on the parts relevant to this paper. These are explained in the sub-sections that follow. Section 3.1 discusses the benefits of tailored backtesting engines for specific research tasks. Section 3.2 is an overview of the expression language that provides the foundation for formalising strategy development processes, with specific examples of decision point formalisation. Section 3.3 delves into strategy generation, emphasising the need for randomness for evaluating the robustness of these decision points within the processes. The multiple nested loops required for this robustness analysis make the problem so computationally expensive that renders existing tools unable to

---

[1]The 'look-ahead' bias results in unrealistically high profits due to accidental misalignment of data that allows decisions to be influenced by future data points yet unavailable.

facilitate such research. We choose Differential Evolution as an example, but our tool allows for a varying set of ML algorithms, facilitating hybridisation for robustness analyses in layers of nesting as shown in Figure 2. This adds to the computational simulation requirements that we plan to implement with our novel software design summarised in this section.

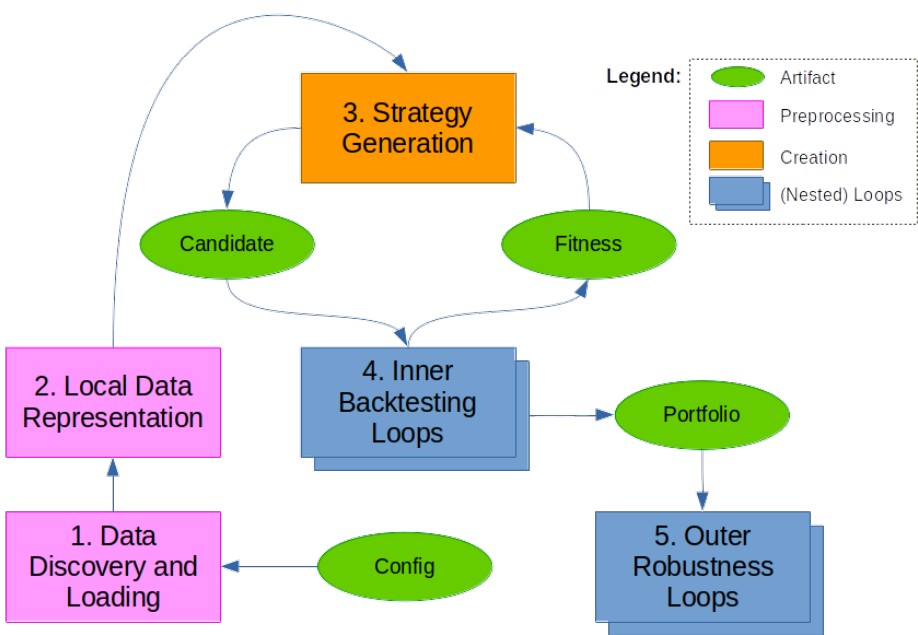

Figure 1: Strategy generation context and architecture of our platform

The platform conducts test scenarios for automated strategy development processes. Strategies are generated for specific markets where individual candidate strategies undergo profitability testing and are then selected to form a *portfolio of strategies* to be traded forward. The robustness of this whole process is ultimately assessed through a Monte Carlo simulation. The platform, as shown in Figure 1, executes this in the following steps:

1. **Data Discovery and Loading** retrieves necessary data from registered data sources based on user-defined test configurations. Depending on the deployment decisions, the data sources can be embedded or accessed via a microservice infrastructure. To optimise performance, data is cached locally in a specialised, low-latency NoSQL database that we designed specifically for this purpose (<anonymised>, 2022d). Test processing can occur locally or parallelised in a distributed grid or cloud computing infrastructure. In the latter scenario, Steps 2 to 4 below are executed on remote computation nodes, while Step 5 is completed on the user's client computer. For this paper, all operations are confined to a single computer. A key technical innovation enables high-performance and low-latency communication through a channel abstraction that creates zero-copy, zero-allocation data pipelines (<anonymised>, 2022c). This expedites test scenario startup times without manual management and copying of large financial data sets.

2. **Local Data Representation** converts loaded data into an optimised in-memory structure, leveraging domain knowledge and ML techniques to extract information and features from the data, as detailed in Section 3.1. Results are stored in primitive double value arrays for indicator blocks (decimal results) and bit sets for signal blocks (boolean results). This approach enables rapid in-memory backtesting, maximising the use of CPU prefetching without relying on slow disk or network data retrievals.

3. **Strategy Generation** constructs strategy candidates by defining and combining expression blocks of indicators and signals. This uses machine learning, specifically differential evolution in this paper, to find local optima in the problem space. Candidate fitness is assessed based on trade frequency, profitability, and other metrics. Utilising our optimised expression language and many semantic shortcuts at this step allows for efficient navigation of the problem space.

4. **Inner Backtesting Loops** assess *individual candidate robustness* using domain-specific heuristics and machine learning, subsequently filtering them into portfolios. These portfolios then undergo an outer walk-forward process (Step 5) that evolves them in regular intervals. Thousands of profitability and success metrics, including average-based Profit/Loss, Sharpe Ratio, and Drawdown, as well as complex statistical tests that assess the entire backtest history, are facilitated with minimal storage and computational overhead. This efficiency is crucial when evaluating hundreds of thousands of generated strategies per second.

5. **Outer Robustness Loops** evaluate the **robustness** of the strategy generation and portfolio selection approach beyond entries & exits and portfolio management.[2] These include risk management, position sizing, and equity curve trading.[3] This allows hypotheses testing on formalised strategy development processes and decisions at higher levels of abstraction. The walk-forward runs of the inner backtesting loops (Step 4) are iterated multiple times here (e.g. 200 times) to Monte Carlo simulate the distribution of out-of-sample results. This *strategy generation is randomised*, yielding slightly different results with each run, based on fluctuations in metrics like Profit/Loss or Sharpe Ratio. Contrasting result distributions allows for comparison and ranking of alternatives. This decision-making process is formalised and fully automated through our domain-specific expression language. We define a **robust strategy development process** as one that has a distribution with **low variance and high average profitability**, does not become unprofitable over many trials, and passes validation checks across multiple markets and time frames (being examples of validation criteria the platform can use). Such quantification of robustness and decision automation (this Step 5) are unique to our platform, surpassing the capabilities of existing platforms and facilitating formal research due to superior processing speed and automation.

Figure 2 shows the different levels of abstraction in this model as ML layers. Hybrid ML approaches can be composed **horizontally** (architectural integration) to create a combined decision on the same layer, **vertically** (data manipulation) by connecting inputs and outputs across multiple layers, or on a **meta optimisation** dimension (model parameter) in one layer (see Anifowose (2020) and Anifowose et al. (2017)).

### 3.1 Backtesting Engines

A fast backtesting engine is the component most critical to performance when applying ML techniques, as numerous alternatives require rapid testing. Three possible types of local data representation are implemented as specialised backtesting engines. These differ mainly in their storage format for simulating trades of a strategy candidate based on historical data:

1. **Historical:** We define 'indicators' as transformations of the price data to extract features and define 'signals' as true/false interpretations of indicators that lead to buy, sell, or hold decisions. This engine stores indicators and signals as in-memory map-based historical caches during live trading to minimise memory usage while providing multithreaded access for chart visualisations (<anonymised>, 2022a). Calculations of these indicators and signals occur lazily through a pull-based interaction, allowing for skipping specific calculations when not immediately required by a strategy.

---

[2]Entries & exits refer to entering a trade and exiting a trade by sending orders to the broker based on rules. Portfolio management refers to selecting markets and strategies regularly based on fitness criteria.

[3]Risk management refers to portfolios, markets, industries, direction, and exposure-based limits. Position sizing refers to rules that are volatility-based or based on equal risk per cent. Equity curve trading refers to temporarily stopping trading after certain losses have accumulated in a day or decoupling a strategy from live execution to simulation during drawdowns in isolated strategy equity.

Figure 2: Machine learning layers

2. **Circular Buffer:** An alternative implementation could employ circular buffers of primitive arrays for in-memory storage of indicators, enhancing calculations for low-latency live trading or optimising backtests of large datasets within a constrained memory space using a moving window. Tick data, with its high volume and smallest granularity, can include multiple 'ticks' (trades) per millisecond, while order book data offers insights into market 'depth' (volume at different prices). For instance, to accelerate portfolio tests on tick data spanning multiple years without fitting into memory, data may be eagerly calculated using an observer pattern. This engine is still under development.

3. **Blackboard:** An alternative approach involves precalculating the entire time series into immutable primitive arrays for indicators and signals, stored in memory following the blackboard pattern (Buschmann et al., 1996, p. 71). This enables memoisation during backtests (Mayfield et al., 1995), with individual backtesting threads able to fetch or add blocks as needed. Unused blocks are automatically evicted based on memory management heuristics like least recently used, soft & weak references, and memory limits. This approach offers rapid access for ML tasks iterating over the time series multiple times to find an optimal variant of a strategy that performs best.

From a computational performance perspective, our focus lies on ML problems employing genetic programming, specifically Differential Evolution, to generate and evaluate numerous strategy candidates. The blackboard option (3) offers the quickest internal representation of this workload. Utilising precomputed primitive arrays as in-memory representations enables extensive CPU prefetching during backtest iterations. Our column-oriented storage is favoured over row-oriented storage to minimise CPU cache misses (Ragan-Kelley, 2014, p.34 ff.). Each column corresponds to an indicator, signal, or another expression that memoises a function on multiple indicators or signals.

## 3.2 Expression Language

The platform uses a domain-specific expression language to formalise *building blocks* for strategy generation (indicators and signals) and to automate portfolio management and other high-level decisions. <anonymised> (2022b) discusses the design and choice to use this expression language in detail. In summary, it is only possible to achieve a high performance by tightly integrating the expression language into

the surrounding context (in our case a trading platform). We also compare our expression language against alternatives in the Java Virtual Machine in that publication. Our implementation is significantly faster while also being more feature-rich. Other contender expression languages are either proprietary and unavailable for comparison or unsuitable because they are written in a different programming language which makes integration impossible or performance too slow. Table 14 extends this analysis by comparing our integrated solution against some available proprietary solutions that also use expression languages. The results show that our generalised concepts can help improve the efficiency of future expression languages and, hence, we offer our expression language implementation as an open source solution.

Indicators (decimals) can be combined into signal blocks (boolean) by comparing them against thresholds or each other in various ways as expressions. The expression language is rule-based with boolean guards that trigger actions: *"Rule := Guard → Action"*. Figure 3 shows examples of such expressions and the generation process:

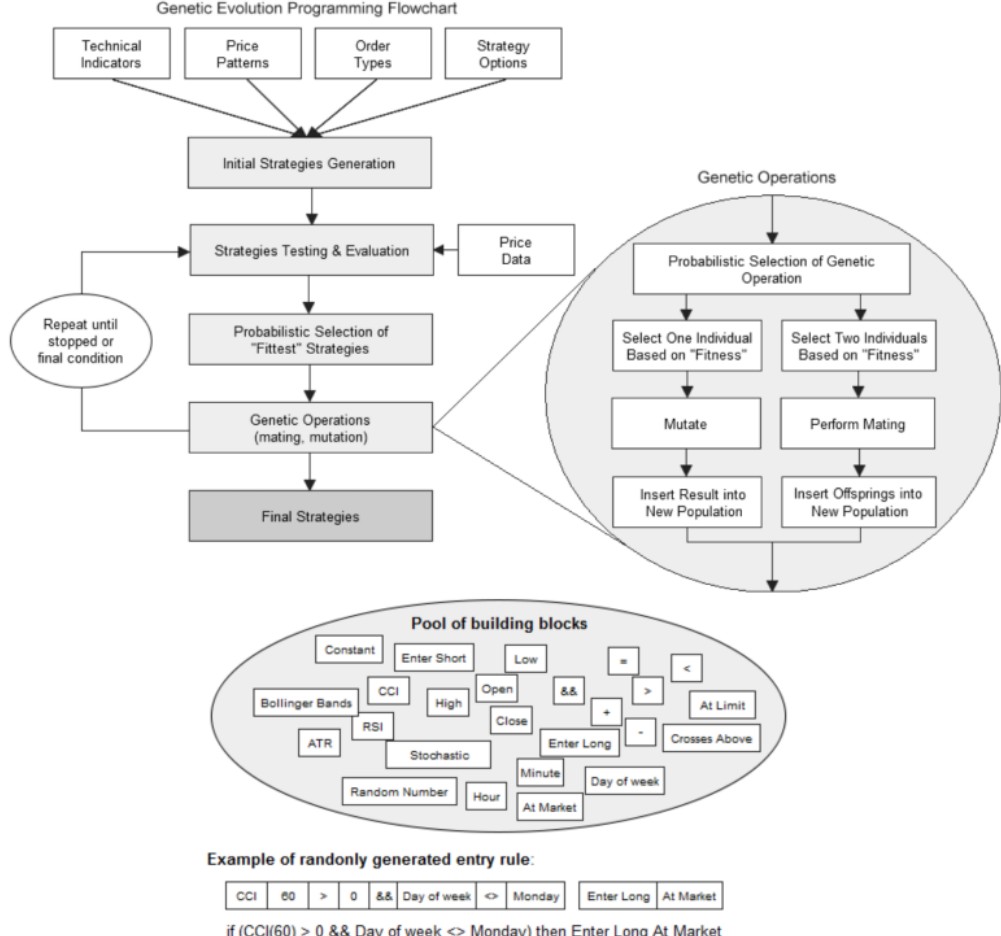

Figure 3: Genetic generation process and boolean expression example (StrategyQuant, 2022)

Like our platform, BuildAlpha (2022) simplifies the strategy generation process by permitting only curated signal blocks. It only allows combining multiple blocks using logical "and" operators. It evaluates simple boolean expressions lazily using shortcutting instead of mathematical expressions. When one condition in a logical "and" combination is false, the remaining conditions can be ignored without calculating additional indicators. This improves the backtesting speed significantly because the signal in a trading strategy is false most of the time. BuildAlpha's backtesting speed is among the fastest available, excluding ours.

### 3.2.1 Expression Language Design

Our language design document (<anonymised>, 2022b) outlines a domain-specific expression language intentionally not Turing complete, prioritising limited resource consumption. It supports variables and functions but excludes features like recursion, loops, arrays, lists, collections, maps, strings, exceptions, declarations, and assignments. Instead, non-recursive functions emulate loops, mappings, decisions, and selections in a simplified fashion that reduces potential coding errors. Functions can be nested arbitrarily as dynamic parameters. Thus, a mathematical view of higher-order functions is supported as nested transformations on *double* series.[4] There is no notion of a functor, function object, or variable for a function in the language. Only the underlying implementation has that representation and is hidden from the user. Instead, the user can think in a series of *doubles* (data) which are provided and transformed by functions to be fed into other functions as parameters. This stems from the fact that the language operates on streamed financial data and can access previous data and calculation results. Since it operates solely on *doubles* and indexes, the function algebra is simplified significantly. The language makes the use of functions and variables transparent to the user by making parentheses optional.[5] Typical mathematical and logical operations are supported and constants are represented as variables. The language is case insensitive and has only a single type *double* for inputs and outputs. Other types like *boolean* are encoded into *double* by seeing numbers above *0* as *TRUE* and all other values as *FALSE. Null* values are represented by the *double* value *NaN* (Not a Number). The underlying implementation may transparently use boolean, Boolean (object), integer, double and generic object types where this provides a performance benefit.

### 3.2.2 Signal Strategy Example

A strategy employing simplified boolean expressions combines multiple signals, which interpret indicators based on domain knowledge, to determine a trade's entry point. Examples of 'domain knowledge' would include technical analysis, statistical measures, or ML models trained on the data. Another simplified boolean expression with an optional time/loss/profit-based stop handles the exit of the trade. Only one trade at a time is allowed. Such a strategy can be expressed schematically in a rule-based form, where a guard is a boolean expression that decides whether to trigger an action or a sequence of actions:

- *Rule := Guard → Action*[6]

- **Entry :=** signal1 && signal2 → **enterLongAtMarket()**

- **Exit :=** with the following Guards and Actions:

  1. signal3 && signal4 → **exitAtMarket()**
  2. stopLoss(volatilityRange1) → **sendStopLossToBroker()**
  3. takeProfit(volatilityRange2) → **sendTakeProfitToBroker()**
  4. maxBars ≥ x → **exitAtMarket()**

An example strategy might take the following form:

- **Entry :=** close[1] < close[0] && volume[1] < volume[0] → **enterLongAtMarket()**

---

[4] For example, a *double* series consisting of {1.1, 1.2, 1.3} is transformed via nested transformations: "*multiply(round(0), 2)*" into a new *double* series consisting of {2, 4, 6}.

[5] This is realised in the syntax. Variables may or may not be suffixed with parentheses. The formats "*variable()*" or "*variable*" are both accepted. Also, parentheses can be omitted for functions with only optional or no parameters. The formats "*function*" and "*function()*" are both acceptable.

[6] The actual syntax uses logical "and"/"or" to combine the guards with mostly implicit actions. This would be the syntax for entry: *"signal1 && signal2 && enterLongAtMarket()"*, and for exit: *"signal3 && signal4 || stopLoss(volatilityRange1) || takeProfit(volatilityRange2) || maxBars ≥ x"*. In the entry, *"&& enterLongAtMarket()"* can be omitted since this would be the implicit action (if configured for the strategy). It is useful to declare it explicitly for actions like limit/stop entries. Limit/stop orders can also be declared for both long and short positions simultaneously by OR combining them into one entry rule. This way, breakout strategies can be expressed. These are implementation details of the language that we express more simply with the rule-based notation.

- **Exit :=** with the following Guards and Actions:
    1. average(close, 10) < average(close, 20) && relativeStrengthIndex(2) < 0.7
       → **exitAtMarket()**
    2. stopLoss(averageTrueRange * 0.5) → **sendStopLossToBroker()**
    3. takeProfit(averageTrueRange * 2) → **sendTakeProfitToBroker()**
    4. maxBars >= 5 → **exitAtMarket()**

An expression is evaluated for each data point in the time series individually in a backtest or as data arrives during live trading. This strategy submits a market order during an upward trend when the last bar closes higher than the previous one (Entry, left) and when there is an increase in volume (Entry, right). Exit occurs when a fast moving average falls below a slow moving average (Exit 1, left) and the Relative Strength Index is below 70% (Exit 1, right). Additionally, trades can exit due to a 50% volatility loss (Exit 2) or a 200% volatility profit (Exit 3). If none of these exit conditions are met, the trade closes 5 bars (e.g., days) after it has been filled by the broker (Exit 4).

Other strategy generators are:

- Mathematical Strategy: Uses mathematical operators to create a decimal value that is compared against a threshold. This requires calculating everything without shortcutting. Since this type of generator is comparatively slow (see Section 5.4), we have not implemented it in our platform.[7]

- Breakout Strategy: This filters trade opportunities based on signals, and then uses a mathematical expression to define a price target for a limit order or a stop order.[8] We have implemented this and can make use of our signal optimisations even though trade simulation is slower than in signal-only strategies.

### 3.2.3   Portfolio Selection Example

Candidate strategies can also be selected for a portfolio through a ranking and filter expression. By portfolio, we mean a collection of dynamic strategies on one or more traded instruments (assets). Typical portfolios only consider Buy & hold investments in stocks, bonds, and funds. The schema might look like this:

- **Rank & Filter :=** with the following Guards and Actions:
    - fitnessFilter1 && fitnessFilter2 → **removeCandidateStrategy()**
    - rankDesc(rankFitness) <= portfolioSize → **removeCandidateStrategy()**

An example expression might look like this:

- **Rank & Filter :=** with the following Guards and Actions:
    - tradesCount >= 100 && sharpeRatio > 0.8 → **removeCandidateStrategy()**
    - rankDesc(profitLoss / maxDrawdown) <= 10 → **removeCandidateStrategy()**

This expression forms a portfolio of the 10 candidates with the highest return-to-risk ratios and have more than 100 trades and a Sharpe ratio above 0.8.

### 3.3   Differential Evolution (DE)

We use DE (Storn & Price, 1997), as a good starting example, to generate our strategy candidates. However, other generators can also be implemented and tested whether they converge faster to viable strategy candidates.

---

[7]We might find ways to accelerate it should we decide to implement it, for example, as a means of exploring whether other profitable strategies could be generated.

[8]An example of a breakout strategy can be schematically defined as: *"filterLong && enterLongAtStop(longPriceLevel + volatility * factor) || filterShort && enterShortAtStop(shortPriceLevel - volatility * factor)"*.

### 3.3.1 Encoding Strategy Candidates

For a strategy that uses simplified boolean expressions with two "and" combined signal blocks for the entry rule and another two "and" combined signal blocks for the exit rule, the problem would be stated as 4 integer variables ranging from -1 to the number of different available blocks per variable. The following example uses 100 blocks. The maximum index is 99 due to array indexing starting at 0, and each array represents a list of domain-specific rules that extract features from market data.

- Index -1 disables this block.

- Indexes 0 to 20 are variants of moving averages and other long-term trend signals.

- Indexes 21 to 35 are volatility signals based on volume or price action.

- Indexes 36 to 70 are short-term momentum rules for mean reversion based on the Relative Strength Index and absolute momentum measures.

- Indexes 71 to 85 are time and session filters to restrict when signals are to be taken.

- Indexes 86 to 99 are Bollinger Bands and other channel-based signals for long-term breakout signals.

The problem space results in the permutations of the variables being multiplied. A strategy candidate selects 4 of the 101 potential blocks (-1 for a disabled block plus rules 0 to 99). The first two variables are used for the entry signal while the last two are for the exit signal. Figure 4 illustrates a two-dimensional problem space, with the third dimension representing the fitness value of negative Profit/Loss aimed for minimisation. The figure displays the convergence point for DE, with the rings at the bottom serving as contours guiding DE towards the minimum. While typical problem spaces resemble more complex terrains with multiple local minima, our 4-variable setup operates in a five-dimensional space, including the fitness value (fifth dimension). Though challenging to visualise, DE adeptly manages higher dimensions by treating them as guiding contour spaces.

### 3.3.2 Vector Movement for Crossover and Mutation

The rule arrays are sorted logically to group similar rules closely, enabling algorithms like DE to leverage directional movements in the contour space. Significant jumps in indexes due to mutations and crossovers yield distinct rules, depicted as individual inverted mountains or craters in the problem space. Conversely, smaller jumps produce similar rules differing mainly in parameterisation for fine-tuning towards local minima. Our generator, on which we base our benchmarks, incorporates 3818 domain-specific rules and includes efficient negation of those that result in twice as many rules (7636). This design allows sufficient space for smaller jumps to select similar rule variants and ample local minima for identifying multiple strategy candidates. To simplify understanding, we focus on 100 rules and illustrate a three-dimensional problem space in our explanations.

Figure 5 shows how DE calculates the vector movements, for each input that we call 'gene', in the problem space using a crossover and mutation formula adapted from Georgioudakis & Plevris (2020).

Figure 6 illustrates how DE is executed. The algorithm commences with an initial population of 500 randomly generated candidates. The scaling factor is a constant that dynamically adjusts based on a random value, moving at a speed determined by the difference between two random candidate values at the same gene. Given our integer values, we include a rounding step to approximate the modified vector indices for new trial candidates to the nearest integer. To ensure the selected signal remains within permissible bounds, we constrain the results between -1 and 99; selecting -1 disables the corresponding block. This approach enables testing simpler strategies with fewer signal blocks than the maximum allowed. In our implementation, the mutation and crossover steps are consolidated into a single operation without creating an interim vector, conserving resources and enhancing algorithmic speed. Distinctively, we deviate from conventional DE implementations by modifying the population first and then collectively evaluating all candidates in a separate step rather than a sequential loop. This introduces additional randomness to the search and facilitates concurrent backtesting depending on the engine.

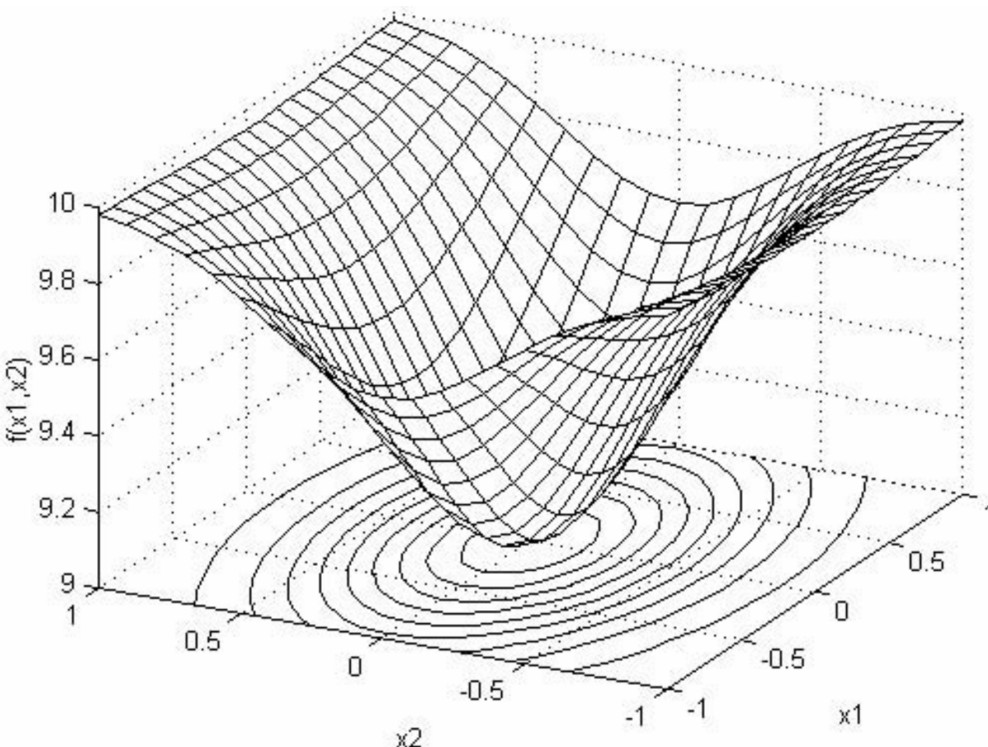

(For better visualisation: Axes of input variables x1 and x2 are normalised between -1 and 1; f(x1,x2) as the fitness output is scaled between 9 and 10)

Figure 4: Two dimensional problem space with fitness as third dimension, adapted from (Price et al., 2005, p. 9)

By sorting the selected blocks within the variables of a generated strategy candidate based on the true count in ascending order, the number of backtests can be minimised. This approach enables the pre-filtering of logically duplicate backtests using a score cache. For new candidates, backtesting is expedited because the initial block typically results in a decision not to initiate a trade, reducing the need to evaluate subsequent blocks. However, this optimisation is rendered ineffective when block compression (see Section 4.2) is activated, as it consolidates blocks into a single "bit set" in advance. Nevertheless, the optimisation remains applicable to expressions that incorporate dynamic blocks.

The parameters for the crossover probability $P$ and the scaling factor $F$ could be meta-optimised separately.

We employ multi-threading to execute strategy generation, with each thread conducting backtests to evaluate the DE algorithm. Each thread iteratively runs DE until the predetermined number of unique and valid candidates is attained. If an excessive number of duplicate or invalid candidates emerge, the process may terminate with fewer valid candidates. Furthermore, strict limits are set on the maximum number of candidates examined, restricting the overall backtests and iterations. DE operates through multiple loops with defined population sizes and convergence parameters dictating crossover and mutation behaviours. Each evolutionary step strives to enhance the population until reaching a maximum iteration threshold or when no improvement occurs over several iterations. The top-performing individual is retained across iterations, and unchanged candidates are not retested. The most outstanding individual from the final population serves as the strategy candidate for subsequent portfolio selection. To prevent redundant backtests across parallel strategy generators, a shared fitness cache is implemented. Duplicate strategies are eliminated from the resultant candidates by comparing final fitness values and trade counts with other candidates. During the generation process, invalid candidates who do not execute trades receive the lowest possible fitness score as a penalty. This approach ensures that DE doesn't prioritise strategies with no trading activity over those that might incur losses.

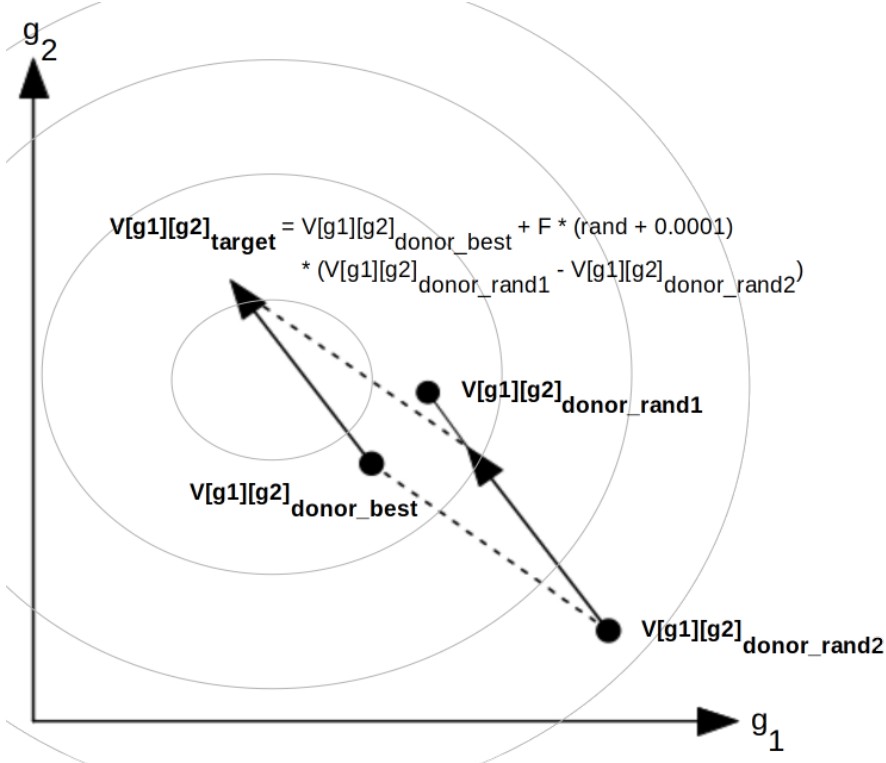

(Instead of x1 and x2 as inputs, we speak of g1 and g2 as genes of the vector tuple V[g1][g2] that represents a candidate that is on the contour space)

Figure 5: Vector movement in the contour space, adapted from (Price et al., 2005, p. 39)

### 3.3.3  Nested Backtest Loops

Figure 7 illustrates various options for implementing nested loops in Post-Optimisation within each backtest concerning strategy generation:

- *Nested Optimisation*: This method employs brute force optimisation, exploring all possible variants for the given blocks to determine the optimal parameters. DE leverages the best fitness value to make decisions about an individual. To control the number of tests, parameters are optimised sequentially, increasing the permutations during the optimisation process. Alternatively, a genetic optimisation algorithm can be employed to minimise tests when dealing with increased permutations.

- *Nested Walk-Forward*: This approach conducts brute force optimisation multiple times in a walk-forward manner, appending the out-of-sample walk-forward steps to calculate the fitness value for an individual.

- *Nested Cross Validation*: This technique divides the testing period into multiple segments. It performs brute force optimisation for each segment individually as the out-of-sample period, while the remaining segments serve as the in-sample optimisation period. The out-of-sample fitness is then averaged across segments for each individual.

There are additional options available for selecting various optimisation parameters for blocks during the initial pre-calculation. These options are referred to as *Pre-Optimisation*, alongside the *No-Optimisation* choice, which retains the default values. *Pre-Optimisation* fully leverages precalculated indicators and signals. While it is more complex than *Post-Optimisation*, the platform calculates all permutations for a block and caches them within the block. This enables values to be reused as primitive arrays throughout optimisation cycles. Although this approach demands a more intensive implementation effort, it facilitates

Initialisation

**1. Generator Template with 100 Signals in 4 Blocks:**
S1[-1..99] && S2[-1..99] && S3[-1..99] && S4[-1..99]
**2. Initialize 500 Candidate Population Vectors:**
$V_1$: S1[0] && S2[10] && S3[58] && S4[-1]
$V_2$: S1[22] && S2[47] && S3[-1] && S4[74]
...
$V_{500}$: S1[75] && S2[11] && S3[67] && S4[99]
**3. Evaluate Fitness of all Candidates**

Differential Evolution

For each candidate "$V_{candidate}$" in population except the best one:

**4. Mutation:**
Pick best candidate "$V_{donor\_best}$" and two distinct random
vectors "$V_{donor\_rand1}$", "$V_{donor\_rand2}$" as donors.

Then create target vector "$V_{target}$" for each Gene "g":
$$V[g]_{target} = V[g]_{donor\_best} + F * (rand + 0.0001) * (V[g]_{donor\_rand1} - V[g]_{donor\_rand2})$$
Where $F$ is the scaling factor with a value of 0.5 and
$rand$ is a uniform random value between 0 and 1.

Then round the gene to the nearest integer and
limit gene values between -1 and 99:
$$V[g]_{target} = limit(round(V[g]_{target}), -1, 99)$$

**5. Crossover:**
Create a new $V_{trial}$ to replace the $V_{candidate}$ in the population:
For each Gene "g" if ($rand < P$) then:
{ $V[g]_{trial} = V[g]_{target}$ } else { $V[g]_{trial} = V[g]_{candidate}$ }
Where $P$ is the crossover probability with a value of 0.5 and
$rand$ is a uniform random value between 0 and 1.

Evaluation

**6. Evaluate Fitness of all Candidates**

Termination

no ← Continue? → yes

X

Figure 6: Differential evolution process

high-performance advanced optimisation workflows, providing DE with a greater level of abstraction for determining strategy candidates. However, for our comparison benchmarks, we deactivate these nested loops since no other platform supports them. Instead, optimisations are streamlined into multiple blocks for the same rule.

The fitness evaluation employs deliberately simple and fast algorithms to swiftly measure specific statistics about strategies. This approach ensures that nested loops execute only the essential calculations required to assess an individual strategy. When a strategy candidate is presented for portfolio selection, it undergoes reevaluation using a more comprehensive set of statistics. This comprehensive suite enables intricate ranking and filtering logic to determine which candidates qualify for portfolio inclusion.

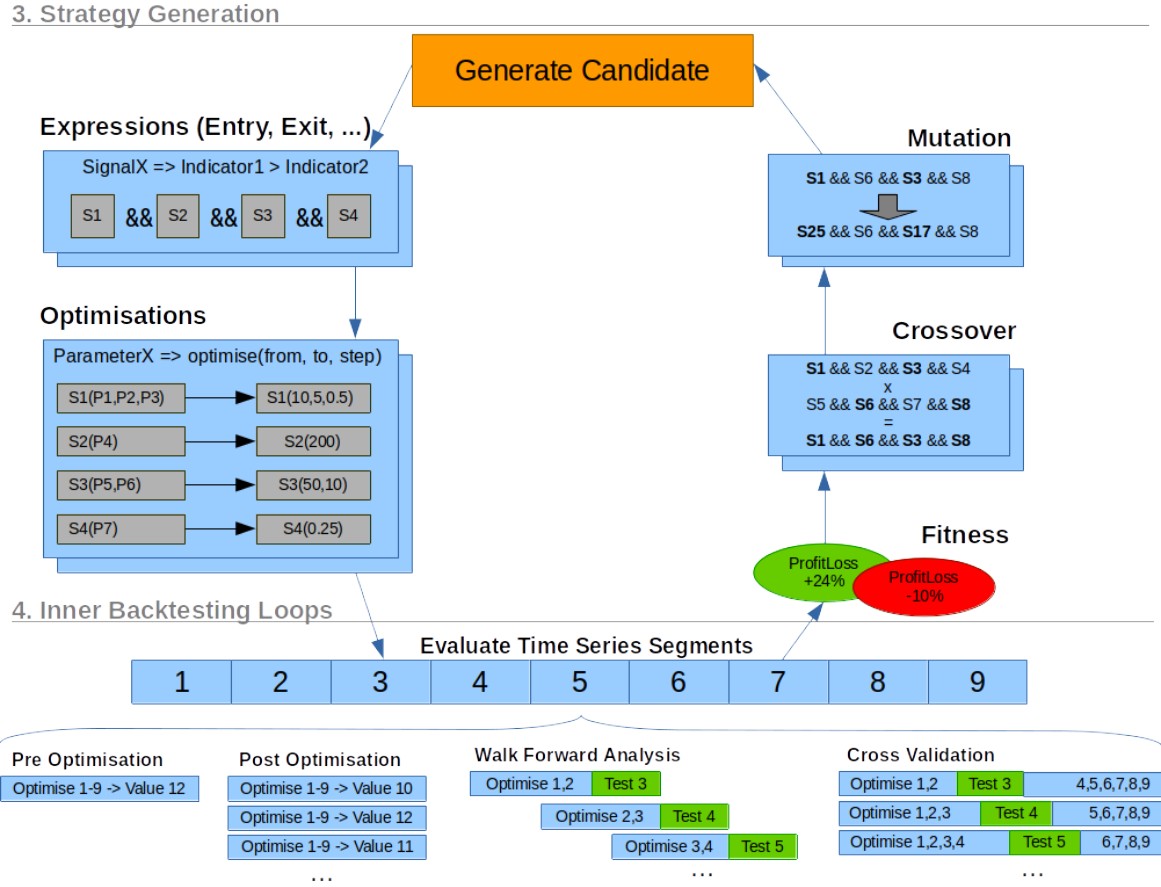

Figure 7: Candidate generation and evaluation process

The advanced statistical suite computes equity-based metrics like Profit/Loss and Sharpe Ratio on a daily basis and feeds this information to fast algorithms calculated eagerly. Similarly, order-related statistics such as Win Percent and Z-Score for Consecutiveness are also pushed eagerly.

More intricate and resource-intensive algorithms, used to validate candidate robustness, are lazily computed based on other statistics, stored daily equity curves, or individual trade events. Portfolio rules can also assess statistical properties like correlation or cointegration between candidates, or employ sophisticated hybrid machine learning techniques at this stage.

By blending eager and lazy calculations, portfolio decisions are expedited for a given configuration. The immutable results can either be stored in files for subsequent analysis using external tools or utilised in the following analysis steps.

## 4 Key Contributors to Performance Improvements

In this section, we summarise the key optimisations undertaken on the platform to achieve performance evaluations that exceed those of extant platforms according to the measures reviewed in Section 5 below.[9] The hardware used comprises an Intel i9-9900k (8 physical cores, 16 virtual cores, 16 MB cache, from 3.6 GHz up to 5 GHz turbo boost) with 64 GB RAM and SSD storage. We use EUR/USD exchange rate as the instrument, with tick data sourced from DukascopyBankSA (2022b) including the bid/ask spread. One

---

[9]These measures are processing speed on ticks and bars for backtesting and optimising classical and ML-generated trading strategies.

minute, four hour, and daily bars are aggregated from this data and used throughout this paper. We next review the performance contributors in order of their importance.

## 4.1 Faster Simplified Boolean Expressions

Our platform leverages similar speed improvement measures as BuildAlpha (2022) while introducing unique optimisations. We store logical signal blocks efficiently using *bit sets*, utilising just 1 bit per value compared to 8 bits in primitive boolean arrays in Java. These bit sets employ primitive long array types in Java, with each bit representing a boolean value, resulting in an eightfold memory reduction and improved CPU prefetching efficiency. Additionally, we utilise bit set "and" operations to **compress** multiple signals into one bit set prior to backtesting. This enables **skipping** unnecessary backtests for bit sets that remain consistently false, determining trade execution only when required. This strategy, termed **"Skipping Compressed Bit Sets,"** optimises the full backtesting logic to execute trades only when necessary, a feature elaborated upon in the benchmarking Section 4.2..

## 4.2 Boolean Compression

Table 1 displays the performance impact of various storage formats for signal blocks, as discussed in Section 3.1. The primary performance enhancement arises from two advanced techniques that we term 'compression and skipping with bit sets'. **Compression** consolidates multiple bit sets using a swift "and" operation, creating a single bit set that facilitates quicker iteration and tracks when the combined expression is true. Entire backtests can be bypassed for false expressions across the entire time series. In the absence of compression, lazy evaluation allows for **skipping unnecessary calculations** through shortcutting of boolean combinations. However, understanding the frequency and locations where the time series is true enables a second optimisation. For sparsely true combinations, skipping pre-computed time series indexes within backtests is achieved by identifying the next true index in the bit set and resuming the backtest evaluation from that point. This optimisation avoids intricate calculations within the backtesting engine, that would have arrived at the same conclusion but with a code path that is more expensive than a bit set lookup.

For automated strategy generation, we opt to create strategies using 8 signal blocks based on simplified boolean expressions for entry, with a time-based exit after one bar. We conduct tests on daily bars spanning approximately 20 years of historical data. The impact of skipping to the next true value during backtests is also demonstrated. We gauge the speed of backtests per second and derive bars per second accordingly.

While compression and skipping optimisations could be applied to boolean lists and arrays, they are likely less efficient and demand considerably more memory than bit sets. A primitive boolean consumes 1 byte, whereas a boolean object in the JVM requires 16 bytes due to the object header. In contrast, a boolean in a bit set uses just 1 bit. Given these inefficiencies, we have not implemented or tested these slower combinations.

Table 1: Boolean storage format performance overhead

| Optimisation | Bars | Backtests/s | Bars/s | Relative |
|---|---|---|---|---|
| RoaringBitmap | 5,839 | 18,439.99 | 107,671,101.60 | -47% |
| BitSet | 5,839 | 34,367.77 | 200,673,409.00 | -1.3% |
| Boolean Array (Primitive) | 5,839 | 34,832.96 | 203,389,653.40 | Baseline |
| Boolean List (Object) | 5,839 | 35,379.05 | 206,578,273.00 | +1.6% |
| Compressed RoaringBitmap | 5,839 | 49,779.17 | 290,660,573.60 | +42.9% |
| Skipping RoaringBitmap | 5,839 | 51,289.57 | 299,479,799.20 | +47.2% |
| Skipping Compressed RoaringBitmap | 5,839 | 104,398.74 | 609,584,242.90 | +199.7% |
| Compressed BitSet | 5,839 | 111,023.99 | 648,269,077.60 | +218.7% |
| Skipping BitSet | 5,839 | 121,204.29 | 707,711,849.30 | +247.9% |
| Skipping Compressed BitSet | 5,839 | 200,746.40 | 1,172,158,230.00 | **+476.3%** |

Table 1 presents a summary of each optimisation's impact, assessed across three computational performance metrics: backtests per second (using generated signal blocks), bars processed per second, and 'Relative' runtime performance based on bars per second, using primitive boolean arrays as the baseline. The tabulated results indicate that leveraging skipping and compressing bit sets **can enhance relative performance by up to 476.3% compared to the baseline** of primitive boolean arrays. RoaringBitmap (Lemire et al., 2017) offers further compression for sparse bit sets by storing only the indexes where the bit set is true, potentially conserving memory for large and sparsely populated data sets. However, this approach incurs a performance penalty of up to 48.0%. We incorporate RoaringBitmap in our platform only when storing over 1 million data points in memory, as the performance impact diminishes to around 20.0% with larger data sets.

## 4.3 Expression Evaluation

In Table 2 we compare the speed of using *cached signal blocks* versus evaluating expressions during backtests. The same scenario is applied as in the above tests. Skipping and compression optimisations are not possible when the expressions are not precalculated and cached into bit sets. We also disable expression simplifications and subexpression elimination for a raw evaluation test. Such simplifications might remove unneeded calculations like "+0" or "-0" while subexpression elimination might turn constant calculations "5+3+2" into a constant value "10". Subexpression eliminations avoid repeating calculations that are already known. This should not be confused with caching or memoisation because there is no additional storage involved. Instead, it is a language feature. When bit sets are not used in favour of the evaluation of expressions, only indicators and price data are retrieved from primitive arrays in a cache that never expires. Because the cache never expires, we call the technique memoisation. Even though the blackboard engine can evict unused data (for example intermediary calculations), the heuristic is smart enough to never evict data that is actively used in our experimental setup.

Table 2: Expression evaluation performance overhead

| Optimisation | Bars | Backtests/s | Bars/s | Relative |
|---|---|---|---|---|
| Raw (Evaluation) | 5,839 | 14,217.35 | 83,015,106.65 | Baseline |
| Subexpression Elimination and Simplification (Evaluation) | 5,839 | 15,730.36 | 91,849,572.04 | +10.6% |
| Skipping Compressed BitSet (Memoisation) | 5,839 | 200,746.40 | 1,172,158,230.00 | **+1,312.0%** |

Table 2 shows the results of unoptimised (raw) and optimised (subexpression elimination and simplification) evaluation against memoisation (with bit sets). As in the previous table, the number of backtests was measured in a multi-threaded strategy generation process. The bars per second and the relative speed differences were calculated based on the unoptimised (raw) evaluation. The tabulated results show that the speed of **evaluations with memoisation and highly optimised bit sets is 1,312.0% faster or 14.1 times as fast** than the evaluation of expressions for every invocation.

The performance of raw evaluation would degrade if the expression blocks contained more extraneous calculations. This scenario would differ with the generation of mathematical or register-based expressions. However, as such generators are not currently implemented in our platform, they aren't tested here. For performance improvements in scenarios where these generators would be beneficial, refer to the performance tests in <anonymised> (2022b). A potential optimisation, not yet implemented but beneficial for generated mathematical expressions, would be eliminating redundant variables or neutral functions in calculations, such as "+a ...-a" or "+f() ...-f()" or "a / a" or "f() / f()".

## 4.4 Memory Requirements

Optimisation techniques like caching or memoisation frequently involve a trade-off between computational speed and memory usage. For our simplified boolean expression strategy generator, memory consumption

in bytes can be determined as outlined below. This is the experimental setup for signal-based strategy generation discussed throughout this document. While bit sets serve as the primary data source for signal-based strategies, we still compute intermediary indicators as primitive arrays of integer, float, double, or long data types. These primitive array indicators may be directly utilised to calculate entry levels for breakout-based strategies.

- **Primitive Array of Double or Long** (timestamp, price, indicator, exchange rate): 5,839 (bars in test) * 8 (bytes) = **46,712 (bytes)**

- **Primitive Array of Integer or Float** (count, index): 5,839 (bars in test) * 4 (bytes) = **23,356 (bytes)**

- **Bit Set of Boolean** (simple signal): 5,839 (bars in test) / 8 (1 bit) = **730 (bytes)**

- **Bars** consist of: 46,712 (start time) + 46,712 (end time) + 46,712 (first tick time) + 46,712 (last tick time) + 46,712 (open) + 46,712 (high) + 46,712 (low) + 46,712 (close) + 46,712 (volume) + 46,712 (mean) + 23,356 (mean count) = 10 * 46,712 + 1 * 23,356 = **490,476 (bytes)**

- **Ticks** consist of: 46,712 (tick time) + 46,712 (ask) + 46,712 (bid) + 46,712 (ask volume) + 46,712 (bid volume) = 5 * 46,712 = **233,560 (bytes)**

- **Indicators for Evaluation** consist of: 141 (double indicators) * 46,712 + 60 (integer indicators) * 23,356 + 119 (boolean indicators) * 739 = **8,075,693 (bytes)**

- **Signal Blocks for Memoisation** consist of: 3818 (boolean signal blocks) * 739 = **2,821,502 (bytes)**

- **Total Memory Usage:** 490,476 (bars) + 233,560 (ticks) + 8,075,693 (indicators) + 2,821,502 (signal blocks) = 8,799,729 (Blackboard Engine) + 2,821,502 (Signal Block Cache) = **11,621,231 (bytes)**

Thus we require about 12 MB of memory for a strategy generator based on about 20 years of daily data. Using primitive boolean arrays instead of bit sets we would require 19,750,514 bytes more, thus about 20 MB of additional memory or about 32 MB in total. **Bit sets save us 63.0% in memory requirements** with this strategy generator. The above calculation can be extrapolated to other time frames. Depending on the markets and their trading session times the amount of data can be significantly less than the example below which is based on a 24-hour trading session. Accordingly, it is better to directly count the given bars in a given time frame of an instrument.

- **Daily** (5,839 bars): **12 MB**

- **4 Hours** (35,034 bars): 12 MB * (24 hours / 4 hours) = 12 MB * 6 = **72 MB**

- **Hourly** (140,136 bars): 12 MB * 24 hours = **288 MB**

- **5 Minutes** (1,681,632 bars): 12 MB * 24 hours * (60 minutes / 5 minutes) = 12 MB * 24 * 12 = **3,456 MB (3.5 GB)**

- **1 Minute** (8,408,160 bars): 12 MB * 24 hours * 60 minutes = **17,280 MB (17.3 GB)**

Nowadays such memory requirements can be easily fulfilled with laptops having 128 GB of available memory (Schenker Technologies, 2022). Reasonably priced servers also provide this amount of memory (HetznerOnlineGmbH, 2022). This ample memory would also facilitate testing multiple instruments concurrently in baskets, employing additional indicators, and caching more signal blocks. During walk-forwards, the entire dataset doesn't need to be loaded into memory. Instead, segments can be individually loaded, allowing, for example, monthly strategy generation using the last three months of tick data. After each segment, prior data is unloaded to accommodate the next in-memory segment. Instruments can also be optimised

separately, loading only one at a time. Investing in more memory could retain segments between runs, accelerating robustness tests. For optimal JVM performance and to prevent significant slowdowns from garbage collection, a rule of thumb would be to allocate to the maximum heap size roughly double the residual memory requirement of the dataset.

The CPU cache may not accommodate all data, even with daily frequency datasets. Our column-oriented primitive array storage maximises CPU prefetching, preloading data from RAM before calculations require it. While virtual memory can potentially hold larger datasets than available hardware memory (RAM), disk swapping can slow backtest speeds due to memory thrashing, as operating systems may not efficiently prioritise memory usage. Memory compression could reduce requirements by up to 40% with 40-70% performance retention (Jennings, 2013). However, it might be more cost-effective to either chunk backtests to fit within hardware limits or increase hardware memory, rather than paying for extended backtest times in cloud computing instances. Thus, we haven't tested these workarounds for larger datasets. Instead, Section 5.6 explores alternative memory allocation techniques, such as memory-mapped files, allowing tests on SSD-stored datasets and leveraging the operating system file cache to decide which memory segments should be cached in RAM. This allows us to perform tests on data sets that are larger than physical memory (RAM).

## 5   Performance Results

In this section, we compare the runtime performance of our Invesdwin platform with competitor platforms. To ensure a fair comparison, we utilise **event-based strategy APIs** from the respective platforms instead of our expression language since some platforms lack specialised expression languages. However, it's worth noting that our expression language serves as a simplified layer over our event-based strategy API, offering identical functionality. For the tests conducted in this section, strategies are manually coded in a feature-rich programming language, avoiding the use of an expression language generated approach.

Our analysis primarily focuses on the raw performance of classical backtesting engines and their optimisation steps. Specifically, we examine differences in storage formats, the potential for multi-threading, and the impact of altering data resolutions on strategy optimisation tasks. In Section 5.4, our attention shifts to performance within the context of a full ML backtesting engine. During this later analysis, we will revisit the role and impact of our expression language.

Before conducting each test, we initiate several warmup runs of backtests on the respective platform. This procedure helps to load necessary data, populate caches, and perform pre-calculations, ensuring more consistent and accurate test results. Each test is repeated 10 times, and the best time achieved is recorded for evaluation purposes.

For inclusion in our comparison, platforms must offer a free demo or trial version to ensure accessibility and fairness. Some platforms were excluded from the evaluation due to challenges in reliably processing tick data or the unavailability of licensed versions suitable for testing purposes. To maintain consistency, we selected backtest durations that would provide a sufficiently large sample of ticks for analysis. Where possible, the same data was imported across platforms to minimise discrepancies. Additionally, we aimed to minimise the gathering of statistics to keep the focus on performance metrics. The primary metric used for our evaluation is the *achieved rate of Ticks per Second (Ticks/s)*.

### 5.1   High Data Resolution: Ticks

Table 3 presents the performance results of processing approximately one year's worth of ticks (discrete data points) for the EUR/USD exchange rates *without* executing trades or strategy logic. This measurement specifically evaluates the *raw performance of processing data points*, excluding the overhead of a transaction or the complexity of strategy logic. To accommodate platforms that do not support ticks as the primary data feed, we also include comparisons using 1-minute bar feeds (aggregated data) spanning three years.

Zipline-Reloaded encountered session alignment issues preventing backtests on 1-minute data. Consequently, we utilised a daily feed and compared it against other Python-based platforms. These findings suggest

Table 3: **Raw performance** in bars or ticks per second (ticks/s)

| Platform | Bars or Ticks | Seconds | Bars/s or Ticks/s | Relative |
|---|---|---|---|---|
| **Daily Data:** | | | | |
| Python: Zipline-Reloaded | 4,528 | 5.098 | 888.19 | -84.5% |
| Python: Backtrader2 | 4,528 | 0.789 | 5,738.91 | *Baseline* |
| Python: PyAlgoTrade | 4,528 | 0.145 | 31,117.59 | +442.2% |
| **Minutes Data:** | | | | |
| Python: Backtrader2 | 1,000,000 | 62.020 | 16,123.83 | -93.0% |
| JForex 4 | 1,052,540 | 46.581 | 22,598.05 | -90.1% |
| Python: PyAlgoTrade | 2,000,000 | 68.263 | 29,298.45 | -87.2% |
| MetaTrader 4 | 1,117,631 | 6.484 | 172,367.52 | -27.8% |
| *Zorro S* | 1,088,716 | 4.750 | 229,203.37 | *Baseline* |
| MetaTrader 5 | 1,117,231 | 0.494 | 2,261,601.21 | +886.7% |
| Invesdwin (Historical) | 1,486,140 | 0.341 | 4,358,181.82 | +1,701.4% |
| Invesdwin (Blackboard) | 1,486,140 | 0.149 | 9,974,093.96 | **+4,251.6%** |
| **Ticks Data:** | | | | |
| JForex 4 | 16,797,607 | 93.778 | 179,120.98 | -91.9% |
| *Zorro S* | 17,479,849 | 7.920 | 2,207,051.64 | *Baseline* |
| MetaTrader 5 | 16,797,607 | 3.098 | 5,422,081.02 | +145.7% |
| Invesdwin (Historical) | 24,232,002 | 3.771 | 6,425,882.26 | +191.1% |
| Invesdwin (Blackboard) | 24,232,002 | 1.727 | 14,031,268.67 | **+535.7%** |

that most purely Python-based platforms, lacking tuned and optimised underlying libraries, demonstrate insufficient speed to remain competitive in our evaluations.

Regarding MetaTrader 4, we couldn't confirm that "processing all ticks" involved genuine ticks rather than interpolated artificial ticks from bars. Hence, we do not categorise this as tick testing. Similarly, Zorro S offers a comparable interpolation mode, which we also exclude from our tests in favour of genuine tick and 1-minute bar configurations available in Zorro S.

For relative comparisons, we use *Zorro S as the baseline* since it also fully loads data into memory, akin to our blackboard engine. Table 3 reveals that our **blackboard engine processes ticks 535.7% faster, or 6.3 as fast as Zorro S**.

Table 4 and its corresponding bar-chart visualisation in Figure 8 show event-based API backtest engines executing trades at tick resolution using a basic (*Moving Average Crossover*) strategy applied to 1-minute bar intervals. Again, we incorporate tests on 1-minute data feeds since not all platforms support tick testing. Additionally, we introduce vector-based backtesting engines for this comparison. Testing vector-based engines on raw data processing speeds is impractical as calculations are necessary for data traversal. Therefore, we evaluate them using this straightforward strategy. The results indicate that our **blackboard engine is 1,303.6% faster, or 14.0 times as fast as Zorro S** when backtesting this simple strategy on ticks.

Generally, Python- and R-based platforms exhibit the weakest performance. Customised platforms like Tradestation, MetaTrader, and Zorro S consistently surpass these platforms. Our platform demonstrates substantial improvements over all these platforms across both minute and tick data. The optimisations detailed in Section 4 are instrumental in achieving this enhanced performance.

Zorro S is noteworthy for its preloading of all data into memory before executing a backtest. Although this accelerates backtesting, it restricts the backtesting window as Zorro was solely available as a 32-bit

Table 4: Performance of **Moving Average Crossover** strategy single backtest (bars or ticks)

| Platform | Bars or Ticks | Seconds | Bars/s or Ticks/s | Relative |
|---|---:|---:|---:|---:|
| **Daily Data:** | | | | |
| a) Python: Zipline-Reloaded | 4,528 | 16.413 | 275.88 | -93.2% |
| b) *Python: Backtrader2* | 4,528 | 1.114 | 4,064.63 | *Baseline* |
| c) Python: PyAlgoTrade | 4,528 | 0.165 | 27,442.42 | +575.1% |
| d) Python: vectorbt | 4,528 | 0.039 | 116,104.56 | +2,756.4% |
| **Minutes Data:** | | | | |
| e) R: quantmod | 200,000 | 20.248 | 9,877.52 | -91.7% |
| f) Python: Backtrader2 | 1,000,000 | 94.977 | 10,528.86 | -91.2% |
| g) JForex 4 (Minutes) | 1,052,540 | 50.696 | 20,763.77 | -82.7% |
| h) Python: PyAlgoTrade | 2,000,000 | 86.709 | 23,065.66 | -80.7% |
| i) TradeStation 9.5 | 6,925,170 | 290.880 | 23,807.65 | -80.1% |
| j) Python: vectorbt | 2,000,000 | 17.257 | 115,895.00 | -3.2% |
| k) *Zorro S* | 1,088,716 | 9.090 | 119,770.73 | *Baseline* |
| l) MetaTrader 4 | 1,117,631 | 6.969 | 160,371.79 | +33.9% |
| m) Matlab | 3,002,266 | 10.917 | 275,008.34 | +129.6% |
| n) NinjaTrader 8 | 370,801 | 1.240 | 299,033.06 | +149.7% |
| o) Julia: Strategems.jl | 3,002,266 | 6.475 | 463,670.42 | +287.1% |
| p) MetaTrader 5 | 1,117,231 | 2.239 | 498,986.60 | +316.6% |
| q) Invesdwin (Historical) | 1,486,140 | 1.763 | 842,960.86 | +603.8% |
| r) Invesdwin (Blackboard) | 1,486,140 | 0.511 | 2,908,297.46 | **+2,328.2%** |
| **Ticks Data:** | | | | |
| s) JForex 4 | 16,797,607 | 102.586 | 163,741.71 | -74.9% |
| t) *Zorro S* | 17,479,849 | 26.780 | 652,720.28 | *Baseline* |
| u) FXCM Trading Station | 2,976,026 | 4.080 | 729,418.13 | +11.7% |
| v) NinjaTrader 8 | 16,472,101 | 8.190 | 2,011,245.54 | +208.3% |
| w) MetaTrader 5 | 16,797,607 | 7.567 | 2,219,850.27 | +240.1% |
| x) Invesdwin (Historical) | 24,232,002 | 6.866 | 3,529,274.98 | +440.7% |
| y) Invesdwin (Blackboard) | 24,232,002 | 2.645 | 9,161,437.43 | **+1,303.6%** |

application at the time of testing. Our evaluations utilised an older licensed version, Zorro S (1.83), where we adjusted the data size to accommodate its backtesting engine limitations.[10]

Zorro shares numerous backtesting features with our Invesdwin platform, offering detailed control over how data feeds are processed during backtests. Our platform, built in Java, fully supports 64-bit memory allocations. Our blackboard engine, which also preloads all data into memory, is solely constrained by available (virtual) memory. The historical engine can test any data volume as it loads data through a moving window from data files.

For a more comprehensive case study, we also implemented the *Workshop 5* strategy from Zorro's documentation (oPgroupGermanyGmbH, 2022a) to compare backtesting speeds in optimisation scenarios. Workshop 5 employs a *Moving Average Crossover* strategy utilising two indicators with a maximum lookback of 500 bars for smoothing filter algorithm normalisation. It initiates trades based on a *crossover* with a specified threshold and employs a volatility-based stop loss.

Trailing stop loss execution is deactivated and identical indicators, algorithms, and settings are used for these tests on both platforms for consistency. In terms of parallelisation, Zorro cannot utilise multiple threads for optimisation backtests, a limitation shared by other platforms in similar scenarios. In contrast, our platform offers broader multi-threading support across various testing scenarios.

---

[10]A 64-bit version of Zorro S (2.50) was subsequently released, supporting larger datasets. Our existing license doesn't cover this version. The changelog mentions minor backtesting speed improvements due to enhanced memory management (oPgroupGermanyGmbH, 2022b). Nevertheless, we anticipate these improvements won't alter the results significantly.

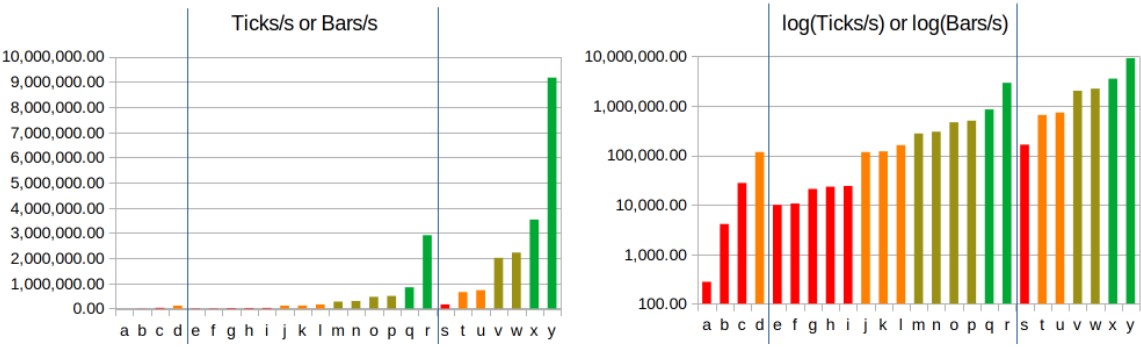

Figure 8: Graphical representation (linear and log y-scale) of Table 4: Performance of **Moving Average Crossover** strategy single backtest (bars or ticks); bars q, r, x and y represent our Invesdwin platform

For equitable comparison, we evaluate both platforms under single-threaded execution while also providing metrics for the Invesdwin platform when multiple threads are employed. Initial backtests operate at tick precision, with the strategy executing trades and logic at four-hour bar intervals. While full statistics are collected in memory during these backtests, report generation time is excluded from the measured duration for both platforms.

It's worth noting that our platform collects a significantly higher volume of statistics than Zorro S, and the time these take is included in the time we record.

First, we compare the single backtest scenario in Table 5. We use the EUR/USD exchange rate during 2015 as the source of the ticks. The results show that our **blackboard engine is 421.3% faster or 5.2 times as fast as Zorro S for backtesting the *Workshop 5* strategy on ticks**.

Table 5: Performance of Workshop 5 **single backtest** (ticks)

| Platform | Ticks | Seconds | Ticks/s | Relative |
|---|---|---|---|---|
| *Zorro S* | 34,443,682 | 53.050 | 649,268.27 | *Baseline* |
| Invesdwin (Historical) | 24,232,002 | 41.946 | 577,695.18 | -11.0% |
| Invesdwin (Historical, Precalculated) | 24,232,002 | 30.605 | 791,766.12 | +21.9% |
| Invesdwin (Blackboard) | 24,232,002 | 7.159 | 3,384,830.56 | **+421.3%** |

The *Precalculated* optimisation utilises file storage to store indicator results, retrieving these values from the file during successive backtests rather than recalculating the indicators each time. Whether stored on a hard disk or SSD, the impact is minimal due to the operating system file cache retaining hot segments in memory. The series comprises only a timestamp and a double value over the entire time range at four-hour intervals (the resolution for strategy decisions), resulting in data small enough (approximately 0.5 MB in size) to fit into memory after an initial run.[11]

The primary advantage stems from eliminating the need to repeatedly calculate complex indicators. This benefit amplifies when more backtests can share these precalculations. This optimisation isn't necessary for the blackboard engine since it loads everything from precalculated primitive arrays in memory.

In Table 6, we present a comparison of a concise optimisation run where three strategy-specific parameters (smoothing filter bars, crossover threshold, volatility multiplier for stop loss) are adjusted in 10% increments for their values, although platforms compute this differently. This serves as a crucial parameter tuning example essential for an ML learning platform. Each parameter undergoes individual optimisation, leading to 32 backtests in our engine. While Zorro performs these backtests sequentially, the Invesdwin platform can

---

[11]If it does not fit for decisions made in higher data resolutions (e.g. ticks), it will still improve results if the calculations that are skipped are costly enough to offset the disk input/output overhead.

concurrently execute multiple backtests in a single thread, allowing data loading pipelines to be reused across multiple backtests. This approach enhances throughput by **reducing file access** compared to conducting each backtest individually.

In environments with multiple cores, Invesdwin distributes backtests among available threads, enabling multiple strategies to operate on the same data stream across the entire time range within a single thread. This methodology boosts throughput, with each thread reusing data pipelines for simultaneous backtests. For the blackboard engine, this optimisation is deactivated as executing each backtest separately in memory proves faster, leveraging improved CPU prefetching due to limited CPU cache sizes. Multi-threading in the blackboard engine still exists, but each thread manages a single strategy. Regardless of the number of concurrent strategies within a thread, once a thread completes its tasks, it proceeds to the subsequent strategy or group of strategies until all necessary backtests conclude.

Since a file buffer cache has been introduced in our NoSQL database it is faster to only run a few (e.g. 5) backtests together per thread in the historical engine, because the overhead of accessing the files is already reduced considerably by the database. The results show that our **blackboard engine outperforms Zorro S by 1,731.0% or 18.3 times as fast in step-wise optimising the *Workshop 5* strategy on ticks**.

Table 6: Performance of Workshop 5 **step-wise optimisation** (ticks)

| Platform | Backtests | Seconds | Backtests/s | Ticks/s | Relative |
|---|---|---|---|---|---|
| *Zorro S* | 38 | 798.210 | 0.048 | 1,639,743.82 | *Baseline* |
| **Invesdwin, 1 Thread:** | | | | | |
| Historical | 32 | 572.197 | 0.056 | 1,355,169.75 | -17.3% |
| Historical, Precalculated | 32 | 427.648 | 0.075 | 1,813,229.72 | +10.5% |
| Blackboard | 32 | 155.639 | 0.206 | 4,982,196.39 | +203.8% |
| **Invesdwin, 12 Threads:** | | | | | |
| Historical | 32 | 112.054 | 0.285 | 6,920,092.67 | +322.0% |
| Historical, Precalculated | 32 | 91.129 | 0.351 | 8,509,081.24 | +418.9% |
| Blackboard | 32 | 25.382 | 1.261 | 30,550,156.17 | **+1,731.0%** |

Table 7 illustrates the *scalability* of the Invesdwin platform when optimising all three parameters simultaneously through a brute-force approach that explores all possible combinations of parameter values. This increases the permutations, demanding a substantially higher number of backtests (1,331 in our study). We maintain multi-threaded execution for these tests. The platform incorporates heuristics to limit the number of parallel backtests, preventing memory exhaustion. If scheduled backtests exceed simultaneous execution capacity, they run in multiple step-wise chunks.

For relative comparison, we employ the same baseline as in Table 6, since Zorro S's backtesting speed doesn't notably alter with increased backtests. The results show that **our blackboard engine is 1,285.6% faster or 13.8 times as fast as Zorro S when brute-force optimising the Workshop 5 strategy on ticks**. In practical terms, this means waiting approximately 24 minutes instead of over 7 hours for 1,331 backtests.

Table 7: Performance of Workshop 5 **brute force optimisation** (ticks)

| Platform | Backtests | Seconds | Backtests/s | Ticks/s | Relative |
|---|---|---|---|---|---|
| **Invesdwin, 12 Threads:** | | | | | |
| Historical | 1331 | 5,478.488 | 0.242 | 5,887,170.81 | +259.0% |
| Historical, Precalculated | 1331 | 3,133.338 | 0.425 | 10,239,429.77 | +524.4% |
| Blackboard | 1331 | 1,419.571 | 0.937 | 22,720,099.71 | **+1,285.6%** |

## 5.2 Medium Data Resolution: One Minute Bars

Usually, there's a trade-off between achieving reliable backtest results with high data resolution and the performance overhead of processing numerous data points. Not all platforms can handle algorithmic trading strategies on ticks or may have limitations on the number of ticks processed in a single backtest. Often, the compromise involves conducting tests on 1-minute bars. To compare, we executed both optimisations on 1-minute bars for the EUR/USD exchange rate from 2003 to 2015, ensuring an adequate sample of bars. Initially, we present single-run comparisons in Table 8. The findings indicate that **our blackboard engine outperforms Zorro S by 7,112.4%, or is 72.1 as fast, when backtesting the *Workshop 5* strategy on 1-minute bars**.

Table 8: Performance of Workshop 5 **single backtest** (one minute bars)

| Platform | Bars | Seconds | Bars/s | Relative |
|---|---|---|---|---|
| *Zorro S* | 6.177.906 | 152.700 | 40,457.80 | *Baseline* |
| Historical | 4,747,531 | 12.738 | 372,706.15 | +821.2% |
| Historical, Precalculated | 4,747,531 | 10.319 | 460,076.65 | +1,037.2% |
| Blackboard | 4,747,531 | 1.627 | 2,917,966.19 | **+7,112.4%** |

Next, we conduct step-wise optimisation of three parameters individually after each other in Table 9. The Invesdwin tests were executed directly with *12 threads.* The results show that our **blackboard engine is 14,364.3% faster or 144.6 times as fast as Zorro S for step-wise optimising the *Workshop 5* strategy on 1-minute bars**.

Table 9: Performance of Workshop 5 **step-wise optimisation** (one minute bars)

| Platform | Backtests | Seconds | Backtests/s | Bars/s | Relative |
|---|---|---|---|---|---|
| *Zorro S* | 38 | 3,719.270 | 0.010 | 63,120.03 | *Baseline* |
| **Invesdwin, 12 Threads:** | | | | | |
| Historical | 32 | 59.123 | 0.541 | 2,569,575.16 | +3,970.9% |
| Historical, Precalculated | 32 | 51.136 | 0.626 | 2,970,920.53 | +4,606.8% |
| Blackboard | 32 | 16.640 | 1.923 | 9,129,867.31 | **+14,364.3%** |

We also repeat the same brute force optimisation of all three parameters together (multiplied permutations) in Table 10. The results show that our **blackboard engine is 16,001.9% faster or 161 times as fast as Zorro S for brute force optimising the *Workshop 5* strategy on 1-minute bars**.

Table 10: Performance of Workshop 5 **brute force optimisation** (one minute bars)

| Platform | Backtests | Seconds | Backtests/s | Bars/s | Relative |
|---|---|---|---|---|---|
| **Invesdwin, 12 Threads:** | | | | | |
| Historical | 1331 | 2,027.233 | 0.656 | 3,117,038.72 | +4,838.3% |
| Historical, Precalculated | 1331 | 1,421.373 | 0.936 | 4,445,675.95 | +6,943.2% |
| Blackboard | 1331 | 621.730 | 2.141 | 10,163,517.54 | **+16,001.9%** |

## 5.3 Low (Minimum) Data Resolution: Four Hour Bars

The strategy operates on decisions made at four-hour intervals, allowing us to test using bars within that timeframe. This data resolution proves most efficient as it reduces the number of data points by collapsing them into intervals. The strategy makes the same decisions as in higher data resolutions but works with bars (intervals) rather than ticks (points). Our platform's backtest results remain the same unless the strategy logic's minimum threshold is surpassed. Other platforms may yield inconsistent or erroneous results at lower

data resolutions, unlike our platform. We sidestep this issue by using the historical spread from ticks when executing trades based on bars. This is done by remembering the related ticks for each bar and using them during trade execution even in lower data resolution tests. We call this feature "Skipping Ticks".[12] This feature requires additional work in our backtesting engine due to handling ticks and bars as separate data streams. Disabling this feature can align our performance with other platforms, reducing backtest times by up to 15% compared to our tests with "Skipping Ticks" (not compared to Zorro S).

First, we present single-run comparisons in Table 11, utilising EUR/USD data from 2003 to 2015 to ensure a sufficient bar sample. **Our blackboard engine demonstrates remarkable efficiency, being 3,534.6% faster or 36.3 times as fast as Zorro S when backtesting the *Workshop 5* strategy on four-hour bars**.

Table 11: Performance of Workshop 5 **single backtest** (four hour bars)

| Platform | Bars | Sec | Bars/s | Relative |
|---|---|---|---|---|
| *Zorro S* | 17,652 | 1.640 | 10,763.41 | *Baseline* |
| Historical | 21,125 | 0.409 | 51,650.37 | +479.9% |
| Hist, Precalc | 21,125 | 0.152 | 138,980.26 | +1,191.2% |
| Blackboard | 21,125 | 0.054 | 391,203.70 | **+3,534.6%** |

Second, we run the step-wise optimisation of 3 parameters in Table 12. The Invesdwin tests are directly performed with *12 threads*. The results show that our **blackboard engine is 14,745.1% faster or 148.4 times as fast as Zorro S for step-wise optimising the *Workshop 5* strategy on four-hour bars**.

Table 12: Performance of Workshop 5 **step-wise optimisation** (four hour bars)

| Platform | Backtests | Seconds | Backtests/s | Bars/s | Relative |
|---|---|---|---|---|---|
| *Zorro S* | 38 | 45.520 | 1.197 | 14,735.85 | *Baseline* |
| **Invesdwin, 12 Threads:** | | | | | |
| Historical | 32 | 2.951 | 10.844 | 229,074.89 | +1,454.5% |
| Historical, Precalculated | 32 | 0.762 | 41.995 | 887,139.11 | +5,920.3% |
| Blackboard | 32 | 0.309 | 103.560 | 2,187,702.26 | **+14,745.1%** |

We also repeat the same brute force optimisation of all 3 parameters in Table 13. The results show that in this configuration our **blackboard engine is 17,319.1% faster or 174.2 times as fast as Zorro S for brute force optimising the *Workshop 5* strategy on four-hour bars**.

Table 13: Workshop 5 **brute force optimisation** (four hour bars)

| Platform | Backtests | Seconds | Backtests/s | Bars/s | Relative |
|---|---|---|---|---|---|
| **Invesdwin, 12 Threads:** | | | | | |
| Historical | 1331 | 226.607 | 5.874 | 124,079.90 | +742.0% |
| Historical, Precalculated | 1331 | 22.423 | 59.359 | 1,253,952.41 | +8,409.5% |
| Blackboard | 1331 | 10.954 | 121.508 | 2,566,859.14 | **+17,319.1%** |

## 5.4 ML Performance using Genetic Programming

Having explored the performance traits of classical backtesting engines, we'll now delve into how their performance stands against specialised machine learning backtesting engines. In this section, we transition from traditional strategy development to AI-generated strategies via genetic programming. While it might

---

[12] A trailing stop loss that uses every tick as the minimum data resolution will have varying results on coarser bars even in our platform. Using a trailing stop that only updates every four hours would make results consistent even if finer resolutions were to be used. So it depends on the strategy design, particularly on when decisions are allowed to be made.

be tempting to assume that faster backtests yield superior strategies, this isn't always the case. Quicker backtesting facilitates the rapid identification of potential strategies within the space defined by the strategy generator. However, evaluating numerous candidate strategies can elevate the risk of overfitting and selection bias, potentially increasing the rate of false negatives (Type II error) by not sufficiently rejecting inadequate strategy candidates. Hence, speed serves only as a component in the strategy generation process.

The other critical facet involves *robustness testing* and portfolio selection, which aim to counteract these biases. While our platform's medium-term objective involves researching these aspects, a comprehensive analysis exceeds this paper's scope. We focus here on our enhancements to backtesting speed, empowering us to investigate the resilience of strategy development methodologies. Enhanced backtesting speed also allows us to dedicate more computational resources to exhaustive robustness examinations.

Table 14 and its visualisation as bar-charts in Figure 9 compare the Invesdwin machine learning backtesting engine with other platforms that have this capability. Given that all platforms can potentially generate endless candidates, capturing the total runtime becomes irrelevant. Instead, we allow the platforms to operate for a set duration to amass a sizable sample, subsequently recording their performance. We gauge performance in *backtests per second* and extrapolate from that to determine processed bars per second, rounding the values for clarity.

This means that not all processed bars might be touched by the platform as some optimisations may skip certain data segments and produce identical backtest results. This, however, allows us to compare normalised performance as bars per second regardless of such optimisations (that differ across platforms). We configure a 100% in-sample period, ensuring the genetic programming algorithm leverages all available bars. To evaluate the overhead associated with genetic programming, we employ varying bar counts. Notably, precalculation and warmup phases are excluded from the evaluation. All platforms leverage multiple threads and can harness the CPU's full capacity.

Both BuildAlpha and Invesdwin generate signal-based strategies with 4 entry and 4 exit blocks combined with identical settings. StrategyQuantX generates signal-based strategies with 4 entry blocks and a time-based exit. Adaptrade and GeneticSystemBuilder generate mathematical entry expressions with 4 indicators compared against a threshold with a time-based exit.

We leave the population size and other genetic programming parameters at their default values in the respective platforms because this should not influence the final number of bars per second. The exact form of the backtests is inconsequential as long as the threads mainly execute backtests. We don't measure the number of candidate strategies created. Instead, we measure how many backtests were executed. Also, the total number of backtests executed is irrelevant because we simply collect a sufficiently large sample (a few minutes) to get the number of backtests executed per second.

For the relative comparison, we use GeneticSystemBuilder (intraday) as the baseline because it performs similar to the tests of the classical backtesting engines in Section 5. Our ML engine is based on the blackboard version of our classical backtesting engines from Section 5. Results in Table 14 and Figure 9 show that **Invesdwin (intraday) is 59,503.8% faster or 596.0 times as fast as GeneticSystemBuilder (intraday)**. **Invesdwin (20 years) is 554.8% faster or 6.5 times as fast as BuildAlpha (20 years)**.

BuildAlpha (bars h–j) was an inspiration in the design of the Invesdwin platform machine learning backtesting engine because it showed that significantly higher backtest speeds are possible with a specialised backtesting engine for machine learning. Other platforms seemingly use classical event-based backtesting engines judging by the measured speeds. Invesdwin makes use of more advanced optimisations that increase the speed further than BuildAlpha, especially *bit set compression and skipping of false indexes.* Both platforms use simplified boolean expressions as the basis for the strategy generator.

In the Invesdwin platform, it is also possible to implement more powerful strategy generators like the mathematical threshold indicator calculation that GeneticSystemBuilder uses or the breakout strategies that StrategyQuantX can generate. These generators will still run magnitudes faster in Invesdwin even though not all optimisations can be fully utilised. This is because other platforms implement only a subset of possible optimisations. Both StrategyQuantX and Invesdwin generate breakout strategies with 4 signal filters and

Table 14: ML performance: strategy generator performance (12 threads)

| Platform | Bars | Backtests/s | Bars/s | Relative |
|---|---|---|---|---|
| ▌a) GeneticSystemBuilder (4 years) | 1,296 | 113,82 | 147,507.40 | -94.0% |
| ▌b) Adaptrade (4 years) | 1,296 | 215.61 | 304,654.44 | -87.5% |
| ▌c) StrategyQuantX (20 years) | 6,164 | 50.33 | 310,266.73 | -87.3% |
| ▌d) Adaptrade (30 years) | 10,336 | 42.16 | 435,750.42 | -82.2% |
| ▌e) StrategyQuantX (intraday) | 6,272,874 | 0.08 | 472,923.58 | -80.6% |
| ▌f) GeneticSystemBuilder (30 years) | 10,336 | 77.70 | 803,108.00 | -67.1% |
| ▌g) *GeneticSystemBuilder (intraday)* | 3,222,007 | 0.76 | 2,443,505.99 | *Baseline* |
| ▌h) BuildAlpha (1 year) | 259 | 143,530.00 | 37,174,270.00 | +1,421.3% |
| ▌i) BuildAlpha (4 years) | 1,098 | 104,920.00 | 115,202,160.00 | +4,614.6% |
| ▌j) BuildAlpha (20 years) | 6,284 | 27,420.00 | 172,307,280.00 | +6,951.6% |
| ▌k) Invesdwin (1 year) | 314 | 916,669.56 | 287,834,241.80 | +11,679.6% |
| ▌l) Invesdwin (4 years) | 1,252 | 596,633.89 | 746,985,630.30 | +30,470.2% |
| ▌m) Invesdwin (20 years) | 5,839 | 193,252.58 | 1,128,401,815.00 | +46,079.6% |
| ▌n) Invesdwin (intraday) | 1,482,424 | 982.46 | 1,456,422,283.00 | **+59,503.8%** |

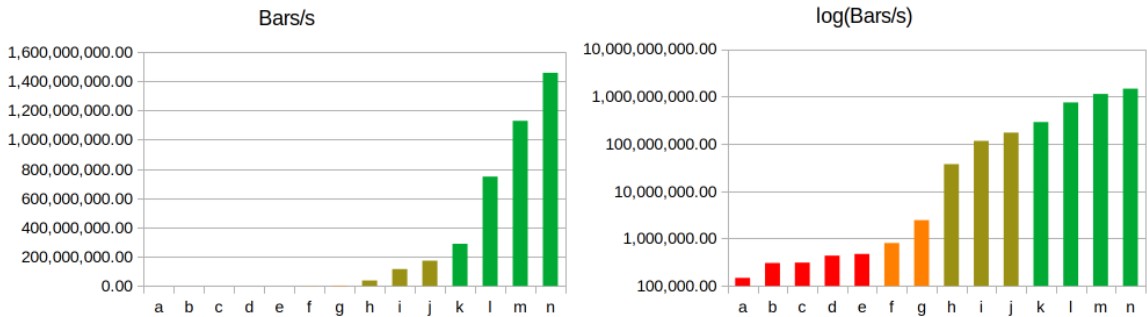

Figure 9: Graphical representation (linear and log y-scale) of Table 14: Strategy generator performance (12 threads); bars k–n represent our Invesdwin platform

a time-based exit in the tests reported below. Table 15 shows that **Invesdwin is 245,399.6% faster or 2,455 times as fast compared to StrategyQuanX for the more sophisticated breakout strategies**.

Table 15: ML performance: breakout generator performance

| Platform | Bars | Backtests/s | Bars/s | Relative |
|---|---|---|---|---|
| StrategyQuantX (Breakout) | 6,164 | 42.35 | 261,073.17 | Baseline |
| Invesdwin (Breakout) | 5,839 | 109,767.72 | 640,933,717.10 | **+245,399.6%** |

## 5.5 Main Contributors to Performance Improvement

To separate the main contributors to the performance improvements, we can pick important measurements of the previous sections and tabulate them for relative improvements. Table 16 and Figure 10 show these improvements measured between each step in relative terms. This is not a comparison based on the same problem, but it allows us to visualise our journey to achieving the backtesting speeds we currently have. Changing from a classical backtesting engine (historical) to an in-memory backtesting engine (blackboard) improved backtesting speeds by a factor of 2.74. Applying multi-threading (12 threads) improved the speed by a factor of 6.13. Switching to our custom expression language allowed us to gain an additional factor of 2.71. Optimising these expressions only gained us a factor of 1.10 in terms of backtests, though switching to

bit sets for the memoisation of calculations improved the speed by a factor of 2.18. Compressing these bit sets gained an additional factor of 3.23 and coupling that with our skipping heuristic led to the final factor of 1.80. The total of all these improvements leads us to a **gain of 646.45 times** in terms of backtesting speed. Taking our performance measurements from the JForex platform as a starting point, we **actually gained about 6543.95 times** in terms of speed.

Table 16: Relative performance contributions

| Platform | Bars/s or Ticks/s | Relative |
|---|---:|---:|
| **JForex:** | | |
| ▌a) 1 Thread (Ticks) | 179,120.98 | -90.1% |
| **Historical Engine:** | | |
| ▌b) 1 Thread (Ticks) | 1,813,229.72 | Origin |
| **Blackboard Engine:** | | |
| ▌c) 1 Thread (Ticks) | 4,982,196.39 | +174.5% |
| ▌d) 12 Threads (Ticks) | 30,550,156.17 | +513.2% |
| **Expression Engine:** | | |
| ▌e) Raw Evaluation (20 years) | 83,015,106.65 | +171.7% |
| ▌f) Optimised Evaluation (20 years) | 91,849,572.04 | +10.6% |
| ▌g) BitSet (20 years) | 200,673,409.00 | +118.5% |
| ▌h) Compressed BitSet (20 years) | 648,269,077.60 | +223.0% |
| ▌i) Skipping Compressed BitSet (20 years) | 1,172,158,230.00 | +80.8% |

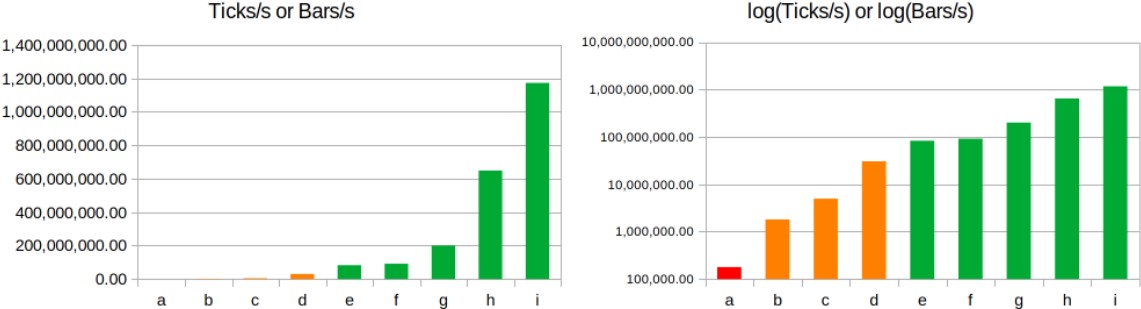

Figure 10: Graphical representation (linear and log y-scale) of Table 16: Relative performance contributions

## 5.6 Scalability Analysis

Scalability comes in two flavours that we can test on a single computer. One is utilising hardware with more CPU cores and another is utilising additional RAM for larger data sets. For the latter we analyse alternative memory allocations with a larger data set; specifically, EUR/USD with 4,370,262 bars in 1-minute intervals from 2003.01.01 to 2015.01.01. Additionally, we test the signal-based strategy generator, as in Table 14, but on better hardware. Table 17 shows the results. Using an almost **3 times larger data set** (4,370,262 instead of 1,482,424 bars as in Table 14 at bar n), our **_OnHeap_ results become 27.6% slower** (1,054,675,328 instead of 1,456,422,283 bars/s as in Table 14 at bar n). This is likely caused by garbage collector overhead.

By using **the memory allocation with memory mapped files we achieve 61.5% to 100.2% faster performance for each hardware we tested on**. *OffHeap* being slightly slower than MemoryMappedFile could be due to pointer compression being disabled for Java processes that are larger than 32 GB in memory. Even if the memory is allocated *OffHeap* and does not count as JVM heap size, we still need to allow the process to have a significantly higher limit than 32 GB of memory which allows the heap to grow beyond 32 GB. This is because the garbage collector is normally greedy and automatically increases the heap size within limits to reduce the frequency of full garbage collection cycles. This is possible as long as the remaining

| Memory Allocation | Backtests/s | Bars/s | Relative (overall) | Relative (per machine) |
|---|---|---|---|---|
| **Intel i9-9900k ↩** | | | | |
| **8-Core (12 threads):** | | | | |
| OnHeap | 241.33 | 1,054,675,328 | Baseline | Baseline |
| OffHeap | 341.06 | 1,490,521,558 | +41.3% | +41.3% |
| MemoryMappedFile | 378.10 | 1,652,396,062 | +56.7% | +56.7% |
| MemoryMappedFile (tmpfs) | 429.40 | 1,876,590,503 | +77.9% | +77.9% |
| **Intel i9-12900HX ↩** | | | | |
| **16-Core (20 threads):** | | | | |
| OnHeap | 671.58 | 2,934,980,554 | +178.3% | Baseline |
| OffHeap | 1,040.18 | 4,545,859,127 | +331.0% | +54.9% |
| MemoryMappedFile | 1,084.39 | 4,739,068,410 | +349.3% | +61.5% |
| MemoryMappedFile (tmpfs) | 1,011.32 | 4,419,733,366 | +319.1% | +50.6% |
| **2x AMD EPYC 7643 ↩** | | | | |
| **48-Core (96 threads):** | | | | |
| OnHeap | 2,380.24 | 10,402,272,423 | +886.3% | Baseline |
| OffHeap | 4,477.99 | 19,569,989,533 | +1,755.5% | +88.1% |
| MemoryMappedFile | 4,423.09 | 19,330,062,150 | +1,732.8% | +85.8% |
| MemoryMappedFile (tmpfs) | 4,766.20 | 20,829,542,744 | +1,875.0% | +100.2% |

Table 17: Signal Strategy Generator Performance on Better Hardware and Alternative Memory Allocations

memory is not fully occupied by the *OffHeap* allocations, such that sufficient space is left for the heap size to grow into.

With memory mapped files we have more control over this and can force the process to use less than 32 GB of memory which allows pointer compression to stay active. This is because the memory mapped files do not count as process memory. Instead, the operating system file cache has control over these memory allocations. All tests were performed on Linux kernels that support the operating system file cache per default. However, this is only relevant for the *MemoryMappedFile* tests where we store the files on a local SSD drive. The *MemoryMappedFile (tmpfs)* tests store the file on a temporary file system (tmpfs) that keeps the files completely in memory. In most cases, tmpfs is faster, but there are cases where the operating system file cache performs better (on Intel i9-12900HX). In most cases, however, memory mapped files (regardless of tmpfs or not) seem to be preferred over *OffHeap* allocations.

**Parallelism.** When comparing the speed improvements that the CPU cores provide, we can see that on the older Intel i9-9900k (12 threads) we achieve up to 154,382,541.00 bars/s per thread. On the newer Intel i9-12900HX (20 threads) we achieve up to 236,953,420.50 bars/s per thread. On the also recent 2x AMD EPYC 7643 48-Core (96 threads) we achieve up to 216,974,403,60 bars/s per thread. Thus ignoring differences in the architecture of the CPUs themselves, it seems like more CPU cores scale more or less linearly the machine learning performance. This trend should be exploitable further in grid computing or cloud infrastructure when more nodes are used. This will also depend on communication overhead which should be negligible because this is an **embarrassingly parallel** computational problem that does not require much communication between nodes. To validate the scalability claim, we reran the above test with the default *OnHeap* memory allocation and measured the performance on the *2x AMD EPYC 7643 48-Core* machine with successively more threads. Table 18 shows the results and Figure 11 visualises the expected scalability (calculated by the theoretical increase based on threads from the baseline) against the actual measured scalability. We can see that the actual curve matches the expected curve closely.

## 6 Conclusions

To address the technical FinTech challenge of testing a vast number of automatically generated trading strategies within a machine learning framework, we developed and evaluated a high-performance compute engine

| Threads | Backtests/s | Bars/s | Relative (actual) | Relative (expected) |
|---------|-------------|--------|-------------------|---------------------|
| 8 | 208.96 | 913,209,947.50 | Baseline | Baseline |
| 12 | 351.90 | 1,537,895,198.00 | +68.4% | +50% |
| 32 | 1,148.46 | 5,019,071,097.00 | +449.6% | +300% |
| 64 | 1,721.36 | 7,522,794,196.00 | +723.8% | +700% |
| 96 | 2,380.24 | 10,402,272,423.00 | +1,039.1% | +1,100% |

Table 18: Signal Strategy Generator OnHeap Threads on 2x AMD EPYC 7643 48-Core

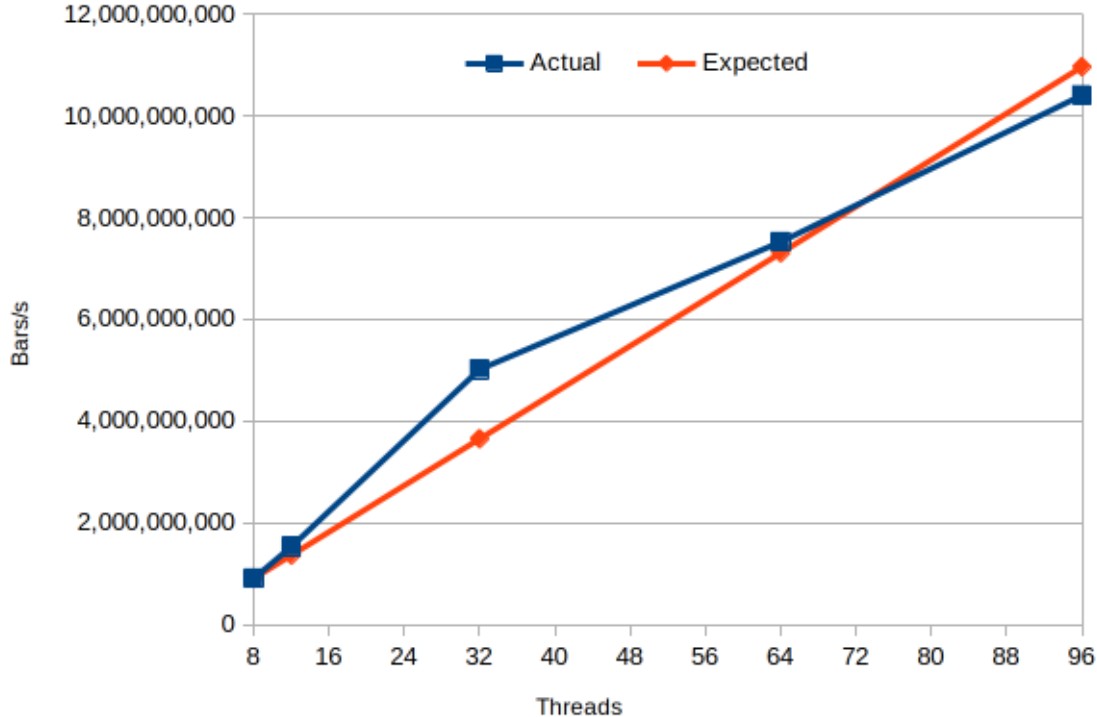

Figure 11: Graphical representation of Table 18: Signal Strategy Generator OnHeap Threads on 2x AMD EPYC 7643 48-Core

within our Invesdwin platform. In this paper, we pinpointed key factors that boost runtime performance, offering developers insights for building similar machine learning engines.

Accelerating backtesting engines is essential for automating strategy development processes. This advancement enables research at a higher abstraction level, addressing specific financial questions by industry experts. In this paper, we delved into the design of the Invesdwin platform and how machine learning concepts can be tested with its compute engine. A pivotal feature of the platform is its domain-specific expression language, facilitating the creation of trading strategies and automated portfolio decisions. The flexibility of this language allows for the formulation of hybrid machine learning methods that, despite posing computational challenges, can be rapidly tested, an aspect we are currently exploring.

This paper presents novel speed and performance enhancements in empirical research, targeting low-latency data access with an *in-memory machine learning engine* support. By **skipping compressed bit sets**, we show that strategy generation speed increases by **5.7 times, reducing memory usage by 63%** for the signal generator. Compared to raw expression evaluation, this technology is 14.1 times as fast (see Table 2 in Section 4). Utilising a *compressed "bit set" format* and *skipping* storage parts based on the dataset and strategy ensures rapid reading from financial databases. Additionally, a custom low-latency **compressed**

**NoSQL database** optimises local data storage. Using memory-mapped files integrated with the flyweight pattern for the allocation of primitive arrays for indicators and signals can double performance on large datasets. This design can surpass hardware memory limitations for extensive data sets.

A comparison of our platform's computational performance with existing contenders demonstrates superior performance. Our in-memory (blackboard) engine for classical event-based trading strategies outpaces all tested platforms. In comparison with **Zorro S**, a platform with some similar features, our 'tick' processing speed is 3.0 (*Zorro Workshop 5*) to 14.0 (*Moving Average Crossover*) times as fast in a single thread depending on the tested strategy. **Step-wise strategy optimisation is 18.3 times as fast on ticks, 144.6 times as fast on 1-minute bars, and 148.4 times as fast on four-hour bars** (measured on *Zorro Workshop 5*). Importantly, trade execution remains consistent across different data resolutions, allowing for the use of coarsest granularity bars, if desired, without sacrificing speed. Consequently, we expect the same results as with tick-level execution but with significantly faster backtesting speed.

Our platform exhibits even larger performance gains compared to platforms that can generate trading strategies while using classical event-based backtesting engines. In the "Invesdwin (intraday)" test, our platform operates **596.0 times as fast as "GeneticSystemBuilder (intraday)"**. In signal strategy tests, our "Invesdwin (20 years)" test reports performance **2,589.6 times as fast as "Adaptrade (30 years)" and 3,636.9 times as fast as "StrategyQuantX (20 years)"**. For breakout strategies within the same time frame, our platform is 2,455.0 times as fast as "StrategyQuantX (20 years)". These results underscore that our compute engine achieves unprecedented performance levels in FinTech.

Even when compared to the fastest contender, "BuildAlpha", which also employs a specialised signal strategy generator, our platform **outpaces "BuildAlpha (20 years) by being 6.5 times as fast in the "Invesdwin (20 years)" test**. Notably, the lack of advanced optimisations in BuildAlpha, such as skipping compressed bit sets, is a primary factor for this performance gap.

Furthermore, our platform distinguishes itself not only in raw speed but also in the ability to formally automate and evaluate the robustness of strategy development processes. In summary, our platform offers the fastest backtesting performance and the most comprehensive automation capabilities for strategy development research, setting a new benchmark in the field.

As a result of the meticulous high level performance tuning of our engine, we managed to execute all tests on a single laptop. Highlighting the current capabilities for leveraging parallelism, our backtesting engine exhibits linear scalability across a moderate number of CPU cores (Section 5.6). The platform harbours additional potential for increased parallelism beyond multi-threading, especially when leveraging grid computing platforms and cloud scaling solutions. A pivotal technical contribution that we identify and which significantly bolsters low latency and high throughput is the integration of native communication channels at the heart of our compute engine.

**Future Work.** Our ongoing research focuses on strategy development processes from a financial perspective. Decision points such as risk management, position sizing, and equity curve trading will shortly be automatable through specialised expressions. This paves the way for exploring and evaluating innovative hybrid ML techniques and formulating resilient investment portfolios. As this research progresses, the platform will be enriched with additional algorithms and testing functionalities. The authors foresee substantial interest in these capabilities from both academic and practitioner circles. The ultimate goal is for this groundbreaking platform to become a widely adopted tool in both academic research and professional settings, enhancing market participants' capacity to explore novel data-driven methodologies in portfolio management.

**Tool and Data Availability.** Interested researchers can get free access to the platform (on github.com) for research purposes by emailing "<anonymised>". The data used in this paper is available at Dukascopy-BankSA (2022a) or through an API with a free demo account at DukascopyBankSA (2022b).

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
