# OpenReview forum: "High-Performance Machine Learning for FinTech"
_TMLR — Rejected by TMLR_

### Review · Reviewer_A8ut · 2024-02-02

**Summary Of Contributions:**

The paper presents a high-performance system for ML in FinTech. The authors
describe the system and its components, highlighting and explaining various
aspects.

**Audience:**

No

**Claims And Evidence:**

Yes

**Requested Changes:**

I do not believe that this paper is suitable for TMLR.

**Strengths And Weaknesses:**

While the paper is interesting, it does not appear to make any contributions to
ML. Most of the paper describes aspects that are not related to ML, such as
different implementations of a storage format and the corresponding performance
evaluations. Some parts of the paper seem to reinvent things from the
literature, such as automated ML, without references. Similarly, it is unclear
why a new expression language is required here and why not an existing one could
have been used.

Further, part of the described system is still under development, according to
the authors.

Altogether, it is hard to see what an ML researcher or practitioner would take
from this paper. The described system is heavily engineered for the specific
purpose it is meant for, and it is unclear whether anything would generalize.

---

### Review · Reviewer_9qwM · 2024-02-09

**Summary Of Contributions:**

This paper discusses their practice of developing a high-performance computing engine for portfolio selection with a large number of candidate strategies.

- An expression language was proposed to formulate the complicated strategies expressively.

- A set of optimizations was introduced to accelerate the execution of these strategies.

- An empirical study was conducted to verify the design and implementation of the system.

**Audience:**

Yes

**Claims And Evidence:**

Yes

**Requested Changes:**

- Include a self-contained introduction to clearly state the background, the problem, the challenge, and the concrete discussion of the paper's main unique contribution(s).

- Update the Context and Background section with a clear categorization.

- Update Section 3 with a clear statement of the problem and better organizations.

- Perform the experiments in a production-level runtime.

**Strengths And Weaknesses:**

- Strengths:

   - This paper discusses a practical problem in FinTech.


- Weaknesses:

  - The introduction is poorly written; there is a lack of a self-contained introduction; the main technique contribution is not clearly stated.

  - The section on context and background lacks good categorization and in-depth discussion.

  - Section 3 lacks clear organization; generally, a reader expects a concrete problem formulation. The current presentation interweaves the problem and solution together.

  - The experiments are not conducted in a production environment, which compromises the credibility of the results.

---

### Review · Reviewer_4DAP · 2024-03-15

**Summary Of Contributions:**

The paper presents a compute engine for a platform for Machine Learning (ML) systems for FinTech developed by the team. The paper discusses some techniques that can help to improve the runtime performance of the developed platform. The paper also performs some experiments in order to evaluate the runtime performance of the proposed compute engine for their platform.

**Audience:**

No

**Claims And Evidence:**

No

**Requested Changes:**

+ The paper needs an aggresive editing to address all the comments I mentioned in the Strengths And Weaknesses section.

**Strengths And Weaknesses:**

I think the paper has various serious issues.

The first issue is that I think the paper is not self-contained. The paper tries to evaluate the computational performance of their developed framework for using ML techniques in the FinTech sector, but the developed framework is not well described in this paper, including the related technologies nor related work. This makes it extremely hard to check the rigorousness of the computational performance evaluation, and to see if the appropriate baselines have been thoroughly compared with.

The second issue is that the problem statement is not described so I feel really confused at all the material presented in the paper. The paper states that it presents new methods for improving the runtime performance of their framework, however, there is no description of the actual problem that results in these new methods so it’s really unclear on the novelty and the rigorousness of the presented methods.

The third issue is that there is no description of related works so it’s unclear how the presented methods is compared to related works.

The fourth issue is that when comparing the runtime performance among different frameworks, I’m not sure if the comparison is comparable as different frameworks use different programming languages and other things, so I don’t think they’re really comparable.

In my opinion, I think this paper is more suited to a journal or conference on software systems rather than at the Transaction on Machine Learning Research.

---

### Review · Reviewer_TJx3 · 2024-03-17

**Summary Of Contributions:**

This paper provides a high-performance compute engine to assess candidate trading strategies in the FinTech sector. This compute engine supports the full pipeline of data discovery/loading, in-memory data representation and feature extraction, large-scale strategy generation (which can be done via genetic programming or ML), and robustness testing. This is supported via development of a domain-specific expression language to process data in a way that is amenable to strategy generation, and via various systems optimizations such as compression/caching/memoization to improve runtime performance. The authors show significant performance improvements against other FinTech platforms.

**Audience:**

No

**Claims And Evidence:**

Yes

**Requested Changes:**

I am not sure this paper is a fit for TMLR. It is likely better suited to a systems venue.

**Strengths And Weaknesses:**

Strengths:
* Real-world system with significant runtime improvements, and end-to-end automated processing from data to decisions.
* Advancements are enabled by a combination of programming languages techniques (domain specific language) and systems optimization techniques.

Weaknesses:
* Fit for TMLR: While the paper makes connections to ML (specifically, its use in the strategy generation step), the main contribution of this paper is in the end-to-end data processing pipeline and compute platform, rather than ML itself.
* While the experiments involve comparisons of the overall system with other FinTech systems, ablation studies are likely needed to evaluate how different parts of the end-to-end pipeline contributed to the improvements in results.
* The paper could benefit from clarity improvements. Notably, it was hard to understand the main contribution of the paper from the abstract/introduction.

---

### Author Response · Authors · 2024-03-28
**Final Revision before Decision**

Thanks a lot for the reviews. We have attempted to incorporate all comments by substantially rewriting, rephrasing, rationalising, and editing the manuscript. In particular, the introduction and motivation (Sections 1 and 2) now highlight the contribution of this paper more clearly.

---

> ### Comment · Reviewer_A8ut · 2024-03-28
>
> Thank you for the revision. I'm still not convinced that there is a contribution beyond engineering this particular system in this particular context -- I don't see what would generalize to other applications.

---

> > ### Author Response · Authors · 2024-04-02
> > **Response to Official Comment on contributions and relevance**
> >
> > In response to previous comments, we would like to point out in terms of contributions and relevance of the paper
> > - we have identified key contributors for performance, and hence derived design principles for *high-performace* ML systems
> > - we quantified these contributors in the context of a concrete ML compute engine for FinTech
> > - we developed these improvements on system level, not specific to one ML technique
> > - our techniques address key challenges in terms of efficient data access and memoisation of data, that are applicable to all data-driven ML techniques
> > We thank thank the reviewers for the feedback and for the consideration of the paper.

---

### Decision · Action_Editor_QKZk · 2024-04-19

**Recommendation:** Reject

**Comment:**

The paper presents a computational engine for portfolio selection and management in the FinTech sector. To enhance runtime performance, the authors introduce a domain-specific expression language and implement various system optimization techniques, resulting a significant improvement in runtime compared to other FinTech platforms.

The paper is generally well written, particularly after the author’s revisions, which provide clearer explanations. The platform presented has practical value in real-world applications.  However, as pointed out by the reviewers, the paper’s primary focus is on system optimization, and its contributions to machine learning are very limited. Given its primary focus on system optimization, it may not align with the scope of TMLR, and therefore, rejection is recommended.

**Audience:**

Not much. The paper focuses on system optimization for a FinTech framework. The contribution to machine learning is very limited. The paper does not seem to fit for TMLR’ audience.

**Claims And Evidence:**

Yes. The claims are supported by clear evidence, especially after the authors provided additional clarification during in revision.